# SemBind: Binding Diffusion Watermarks to Semantics Against Black-Box Forgery Attacks

**Xin Zhang** [* 1 2] **Zijin Yang** [* 1 2] **Kejiang Chen** [1 2] **Linfeng Ma** [1 2] **Weiming Zhang** [1 2] **Nenghai Yu** [1 2]

## Abstract

Latent-based watermarks, embedded during the generation process of latent diffusion models (LDMs), facilitate detection and attribution of generated images. However, recent black-box forgery attacks can implant a provider's watermark into images not generated by that provider, using at least one watermarked image and black-box access to the model, thereby undermining provenance and trust. We propose SemBind, the first defense framework for latent-based watermarks against black-box forgery, which binds latent watermark signals to image semantics through a contrastively trained semantic masker. The masker produces near-invariant codes for semantically matched prompts and near-orthogonal codes across different prompts, enabling SemBind to modulate the target latent before applying standard latent-based watermarking schemes. Across four mainstream latent-based watermarking methods, SemBind substantially reduces false acceptance under black-box forgery while preserving image quality and offering a tunable robustness–security trade-off via a simple mask-ratio parameter. Code is available at https://github.com/XinZhang1999/SemBind.

## 1. Introduction

Latent diffusion models (LDMs) (Ho et al., 2020; Sohl-Dickstein et al., 2015; Song & Ermon, 2019) now generate images that are virtually indistinguishable from real photographs, enabling a wide range of creative and assistive applications. At the same time, such realism raises acute concerns about misleading content and deepfakes (Europol

---

[*]Equal contribution [1]School of Cyber Science and Technology, University of Science and Technology of China, Anhui, China [2]Anhui Province Key Laboratory of Digital Security, Anhui, China. Correspondence to: Kejiang Chen <chenkj@ustc.edu.cn>.

*Proceedings of the 43rd International Conference on Machine Learning*, Seoul, South Korea. PMLR 306, 2026. Copyright 2026 by the author(s).

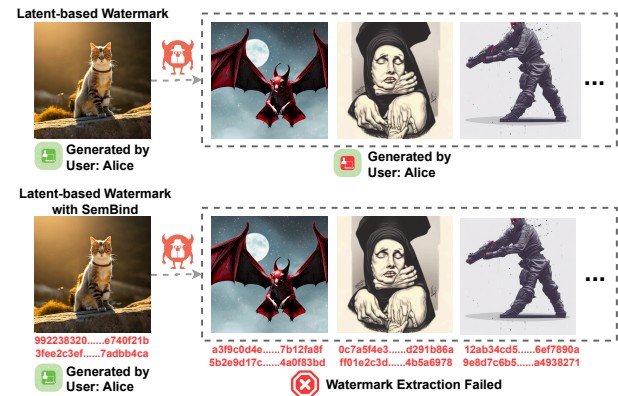

*Figure 1.* Black-box forgery attack and SemBind overview. Latent-based watermarking embeds a pattern in the initial latent noise, which a black-box attacker can transfer to forged images from at least one watermarked example. SemBind additionally binds the latent watermark to a semantic bitstring, causing verification to fail when the forged semantics deviate from the original.

Innovation Lab, 2022; Goldstein & Grossman, 2021), which can be used to deceive individuals, sway public opinion, and facilitate fraud.

Watermarking aims to mitigate these risks by embedding information into generated images for later copyright authentication and provenance tracking. It is already being piloted by governments (The White House, 2023; European Parliament and Council of the European Union, 2024) and major AI providers (Bartz & Hu, 2023; Clegg, 2024) as a key mechanism for responsible deployment.

Watermarking for diffusion models falls into three broad categories. Post-processing methods (Cox et al., 2008; Zhang et al., 2019) modify robust image features after generation, while fine-tuning methods (Fernandez et al., 2023; Xiong et al., 2023; Zhao et al., 2023) amalgamate the watermark embedding process with the image generation process. Latent-based schemes (Wen et al., 2023; Yang et al., 2024b; Gunn et al.; Yang et al., 2025) instead encode a pattern in the initial noise and recover it by inverting the denoising process. This design keeps the diffusion model unchanged, is typically more robust to image transformations, and, more importantly, has provable undetectability (Christ et al., 2024). In particular, undetectability implies that the watermarked outputs are statistically indistinguishable from

non-watermarked ones, and thus the watermark does not introduce a systematic degradation in generation quality.

However, latent-based watermarks are highly susceptible to black-box forgery attacks (Müller et al., 2025; Jain et al., 2025). As illustrated in Figure 1, an attacker with only black-box access and at least one single watermarked image can transfer the watermark to large volumes of illicit content, undermining both watermark owners and AI service providers.

Within this latent-based paradigm, there is no effective defense against black-box forgery. Naïve countermeasures such as tightening detection thresholds (Müller et al., 2025) are largely ineffective: since the watermark lives purely in the initial latent, both natural perturbations and forgery operations only manifest as modifications to this latent, making them hard to distinguish at verification time. As robustness is improved to tolerate more natural perturbations, the scheme simultaneously becomes more tolerant to forged latents and thus less resistant to black-box forgery. *A key motivation of this work is to strengthen latent-based watermarks against black-box forgery without sacrificing their key advantage: for schemes that admit provable undetectability, we preserve the same guarantee under the same setting.*

In this work, we propose *SemBind*, the first defense framework for latent-based watermarks that resists black-box forgery by binding latent watermark signals to image semantics via a learned semantic masker. The masker is trained contrastively to produce near-invariant codes for images from the same prompt and near-orthogonal codes across prompts. During watermarking, SemBind generates an auxiliary clean image for the target prompt, extracts a semantic code, expands and permutes it under a secret key, and uses the resulting mask to multiplicatively modulate the watermarked latent produced by any standard scheme. This design preserves image quality, while a single *mask-ratio* parameter controls the trade-off between anti-forgery strength and robustness to natural distortions.

We validate SemBind on four representative latent-based schemes—Tree-Ring (Wen et al., 2023), Gaussian Shading (Yang et al., 2024b), PRC (Gunn et al.), and Gaussian Shading++ (Yang et al., 2025), by instantiating SemBind-enabled variants for each. Our evaluation covers robustness to common perturbations, resistance to imprinting and reprompting attacks, and image quality and semantic alignment measured by FID (Heusel et al., 2017) and CLIP scores (Radford et al., 2021). Across all four schemes, SemBind substantially reduces false acceptance under black-box forgery while preserving watermark robustness and keeping FID and CLIP essentially unchanged, yielding a controllable robustness–security trade-off via the mask ratio. Moreover, for base schemes that admit provable undetectability, we theoretically prove that SemBind preserves the same unde-

tectability guarantee under the same setting.

In summary, we make the following contributions:

- We propose SemBind, the first defense for latent-based diffusion watermarks against black-box forgery. By learning a semantic masker via contrastive learning and introducing a mask-ratio parameter, SemBind binds latent signals to image semantics, providing strong resistance to forgery while enabling a controllable trade-off with watermark robustness.

- We instantiate SemBind on four representative latent-based schemes. For schemes that admit provable undetectability, we theoretically prove that the SemBind-enabled variants preserve the same undetectability guarantee under the same setting, and empirically confirm that FID and CLIP remain on par with the original baselines.

- We evaluate SemBind under common perturbations and two canonical black-box forgery strategies (imprinting and reprompting), showing substantially reduced false acceptance while preserving robustness and enabling a tunable robustness–security trade-off.

## 2. Related Work

### 2.1. Diffusion Models and Inverse DDIM

Diffusion models synthesize images by iteratively denoising a latent variable that is initially drawn from a Gaussian prior. In latent diffusion models (LDMs) (Rombach et al., 2022), the diffusion process operates in a latent space $\mathcal{Z}$. An encoder $\mathcal{E}$ maps an image $x \in \mathbb{R}^{H \times W \times 3}$ to its latent representation $z_0 = \mathcal{E}(x) \in \mathbb{R}^{h \times w \times c}$, and a decoder $\mathcal{D}$ reconstructs the image as $x = \mathcal{D}(z_0)$. A pretrained LDM therefore consists of the tuple $\Theta = (\mathcal{E}, u, \mathcal{D})$, where $u$ denotes the noise-prediction network (UNet).

Starting from an initial latent $z_T \sim \mathcal{N}(0, I)$, DDIM sampling (Ho et al., 2020) runs a deterministic denoising trajectory: at each step $t$ it uses the trained noise predictor $u(z_t, t, C)$ and the noise schedule $\{\alpha_t\}$ to update $z_t$ until reaching a clean latent $z_0$. We denote the full forward denoising process that maps $z_T$ to $z_0$ by $z_0 = \mathcal{G}_{T \to 0}(z_T; u)$.

Conversely, inverse DDIM (Mokady et al., 2023) approximately retraces this trajectory in reverse: given an image latent $z_0$, it iteratively adds noise using the same predictor $u$ and schedules to obtain an estimate $\hat{z}_T$ of the initial noise, which we write compactly as $\hat{z}_T = \mathcal{I}_{0 \to T}(z_0; u)$.

### 2.2. Latent-based Watermark

In this work we focus on *latent-based* watermarking schemes for diffusion models (Wen et al., 2023; Yang et al.,

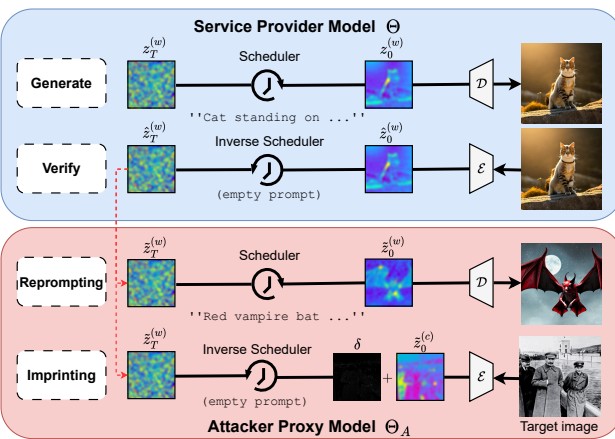

*Figure 2.* Latent-based watermarking and black-box attacks. The adversary inverts a watermarked image with a proxy model $\Theta_A$ to estimate $\tilde{z}_T^{(w)}$. Imprinting optimizes a cover latent so that $\mathcal{I}_{0\to T}(\tilde{z}_0^{(c)} + \delta; u_A) \approx \tilde{z}_T^{(w)}$, while reprompting directly reuses $\tilde{z}_T^{(w)}$ with a new prompt.

2024b; Gunn et al.; Yang et al., 2025; Christ et al., 2024). As illustrated in Figure 2, these schemes encode a message into the initial noise $z_T^{(w)}$ during *Generate*, then approximately recover it by running an inversion scheduler from the generated image during *Verify*. This latent-space design keeps the diffusion backbone unchanged and is compatible with provably undetectable constructions.

We focus on four representative latent-based schemes: Tree-Ring (Wen et al., 2023), Gaussian Shading (Yang et al., 2024b), PRC (Gunn et al.), and Gaussian Shading++ (Yang et al., 2025). Tree-Ring embeds a *zero-bit* watermark by imposing a characteristic pattern in the initial latent, and does not provide a provable undetectability guarantee. Gaussian Shading embeds *multi-bit* watermarks by constraining the initial latent with a secret key and verifying whether the inverted latent remains consistent with this constraint; it admits provable undetectability in the *single-sample* setting. PRC watermarking encodes messages into key-dependent pseudorandom code patterns in the latent and decodes them after inversion, providing *multi-bit* watermarking with provable undetectability in the *multi-sample* setting. Gaussian Shading++ combines PRC-protected seed with GS-style payload, providing *multi-bit* watermarking with provable undetectability in the *multi-sample* setting.

## 2.3. Black-Box Forgery Attack and Threat Model

**Black-box forgery attacks.** Black-box forgery attacks (Müller et al., 2025) are substantially more powerful than earlier "average" template attacks (Yang et al., 2024a), which estimate a fixed watermark pattern by aggregating many watermarked samples. In practice, average attacks are largely ineffective against most latent-based watermarking

schemes, whose watermark signals are instance-dependent and cannot be reliably recovered by simple averaging.

More recent black-box forgery attacks can be broadly viewed as either inversion/optimization-based or distribution-training-based. The imprinting and re-prompting attacks of Müller *et al.* (Müller et al., 2025) use one or more watermarked examples and a proxy diffusion model to transfer the provider's watermark signal to images with different semantics. Jain *et al.* (Jain et al., 2025) improve the computational efficiency of this attack by simplifying the optimization procedure, but this efficiency gain comes at the cost of reduced image quality and lower forgery success rates. WMCopier (Dong et al., 2025) trains an attacker-side diffusion model from a set of watermarked images to imitate the watermark distribution and forge watermarks on arbitrary images. These attacks highlight different practical routes to watermark forgery. We therefore adopt the stronger imprinting and reprompting attacks of Müller *et al.* as the main black-box forgery threat model, and further evaluate SemBind against WMCopier in Appendix G.

**Threat model.** As shown in Figure 2, the service provider holds a private watermarked image generation model $\Theta$, which internally uses a *Generate* procedure to produce watermarked images and a *Verify* procedure to extract the watermark. The attacker's goal is to produce images that are not generated by the provider, possibly with unauthorized or malicious semantics, but can still be falsely verified as carrying the provider's watermark.

We consider a black-box attacker that can query $\Theta$ with arbitrary prompts, obtain at least one or even a dataset of watermarked images, and knows the watermarking algorithm, hyperparameters, and the public design of SemBind following Kerckhoffs' principle. The attacker also uses a proxy diffusion model $\Theta_A$, instantiated either as the same backbone as $\Theta$ in the "match" case or as a weaker public model in the "mismatch" case. However, the attacker has no access to the provider's secret keys, model parameters, gradients, or intermediate latents during embedding.

The detailed attack procedure is shown in Figure 2. The attacker first inverts a watermarked image with $\Theta_A$ to estimate the provider's watermarked initial latent $\tilde{z}_T^{(w)}$. In the *imprinting* attack, the attacker optimizes a small perturbation on a target cover image so that its inverted latent matches $\tilde{z}_T^{(w)}$. In the *reprompting* attack, the attacker directly reuses $\tilde{z}_T^{(w)}$ as the initial noise and samples with a different, potentially malicious prompt, producing new images that still carry the provider's watermark.

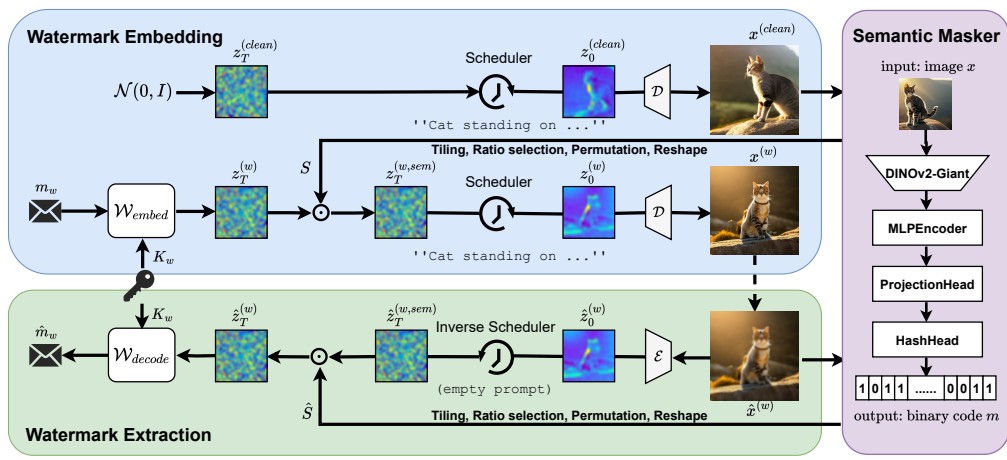

*Figure 3.* The framework of SemBind, including three components: semantic masker, embedding procedure, and extraction procedure.

## 3. Method

Figure 3 gives an overview of SemBind, consisting of three components: the semantic masker, the watermark embedding procedure, and the extraction procedure, which we describe in detail below.

Our central idea is to bind latent-based watermarks to image semantics. During embedding, we map an image to a compact binary semantic code, expand it into a latent mask, and use this mask to modulate the watermarked initial latent. During verification, we recompute the code from the watermarked image and rebuild the mask: if the semantics are preserved, then the masks align, and watermark extraction is only slightly perturbed; if the semantics change, the mismatch induces a strong perturbation that makes forged images fail the watermark check.

### 3.1. Semantic Masker

We introduce a semantic masker $f_\theta$ that maps an image $x$ to a binary code: $m = f_\theta(x) \in \{0, 1\}^B$. This semantic masker is trained and kept private by the service provider.

Two properties are satisfied: (i) images generated from the *same* prompt yield near-invariant codes (small Hamming distance), and (ii) images from *different* prompts yield codes that are approximately orthogonal in expectation (Hamming distance $\approx B/2$).

**Architecture.** The masker $f_\theta$ couples a frozen semantic image encoder with a lightweight MLP-based hashing network. Given an image $x$, a pretrained vision encoder produces a global embedding $e \in \mathbb{R}^{d_e}$ (e.g., the CLS token of a ViT-style backbone). This embedding is then processed by three MLP modules: (i) an encoder $\mathrm{Enc} : \mathbb{R}^{d_e} \to \mathbb{R}^H$ composed of residual MLP blocks with batch normalization and GELU; (ii) a projection head $\mathrm{Proj} : \mathbb{R}^H \to \mathbb{R}^D$

that outputs an $\ell_2$-normalized representation for contrastive learning; and (iii) a hash head $\mathrm{Hash} : \mathbb{R}^D \to \mathbb{R}^B$ implemented as residual fully connected blocks followed by a linear layer that outputs $B$ logits.

The semantic masker $f_\theta$ operates in two modes: a *logit mode* used during training and a *binary mode* used at inference time. In *logit mode*, given a global embedding $e$, the network outputs a hash logit: $\ell = \mathrm{Hash}(\mathrm{Proj}(\mathrm{Enc}(e))) \in \mathbb{R}^B$.

In *binary mode*, we further map $\ell$ to a soft binary code: $b = \tanh(s\,\ell) \in [-1, 1]^B$, and then binarize it into a binary code: $m = \frac{\mathrm{sign}(b)+1}{2} \in \{0, 1\}^B$, where $s > 0$ controls the sharpness of the soft sign.

**Training.** We train $f_\theta$ on a large prompt-conditioned corpus of semantic image embeddings. During training, $f_\theta$ operates in *logit mode*, outputting hash logits for each embedding. Inspired by the cluster-then-quantize paradigm widely adopted in image hashing (Wang et al., 2023; Shen et al., 2024; Wei et al., 2024), we also follow a two-stage routine: first learning a compact and clusterable representation space, and then quantizing it into binary codes.

In **stage-1**, we optimize $\mathrm{Enc}$ and $\mathrm{Proj}$ using a supervised contrastive loss on the normalized features $z_i \in \mathbb{R}^D$. The motivation of this stage is to cluster the raw embedding $e$ produced by a pretrained vision encoder, which often exhibits insufficient compact regularization for downstream binary coding. Although these features are high-dimensional, their *intrinsic* information dimension can be much lower, and the representation space is not explicitly constrained to be compact, leaving substantial redundancy (Zhang et al., 2025). Concretely, $\mathrm{Enc}$ and $\mathrm{Proj}$ map $e_i$ into a compact hyperspherical space and encourage prompt-level clustering.

Given a mini-batch $\{(x_i, y_i)\}_{i=1}^N$ with prompt labels $y_i$ and features $z_i = \mathrm{Proj}(\mathrm{Enc}(e_i))$, we define for each anchor $i$ the set of positives $P(i) = \{\, p \neq i \mid y_p = y_i \,\}$. The **stage-1**

objective is a standard supervised contrastive loss (Khosla et al., 2020):

$$\mathcal{L}_{\text{sup}} = -\frac{1}{N} \sum_{i=1}^{N} \frac{1}{|P(i)|} \sum_{p \in P(i)} \log \frac{\exp(z_i^\top z_p / \tau)}{\sum_{a \neq i} \exp(z_i^\top z_a / \tau)}, \tag{1}$$

where $\tau > 0$ is a temperature. This loss encourages features from the same prompt to cluster on the unit sphere while separating different prompts.

In **stage-2**, we freeze the semantic image encoder and $\text{Enc}+\text{Proj}$, and train the hash head $\text{Hash}$ to *quantize* the learned spherical features into binary codes with desirable bit-space properties. For each sample we compute logits $\ell_i$, codes $b_i = \tanh(s \, \ell_i)$, and use a supervised contrastive loss in the code space,

$$\mathcal{L}_{\text{hash}} = -\frac{1}{N} \sum_{i=1}^{N} \frac{1}{|P(i)|} \sum_{p \in P(i)} \log \frac{\exp\big((b_i^\top b_p / B)/\tau_h\big)}{\sum_{a \neq i} \exp\big((b_i^\top b_a / B)/\tau_h\big)}, \tag{2}$$

where $\tau_h > 0$ is a temperature in the hash space.

In addition, we add three regularizers on $\{b_i\}$: (i) a *quantization* term $\mathcal{L}_{\text{q}} = \mathbb{E}[1 - |b_i|]$ encouraging $|b_{i,k}| \to 1$, (ii) a *bit-balance* term $\mathcal{L}_{\text{bal}} = \frac{1}{B} \sum_{k=1}^{B} (\frac{1}{N} \sum_i b_{i,k})^2$ pushing each bit to have zero mean across the batch, and (iii) a *decorrelation* term $\mathcal{L}_{\text{dcr}} = \|C - I\|_F^2$, where $C$ is the sample covariance of $\{b_i\}$. To further stabilize the codes, we generate two jittered views of each feature and penalize their discrepancy via a bit-consistency loss $\mathcal{L}_{\text{cons}} = \mathbb{E}[\|b_i^{(1)} - b_i^{(2)}\|_1]$.

The overall **stage-2** objective is as follows:

$$\mathcal{L} = \mathcal{L}_{\text{hash}} + \lambda_{\text{q}} \mathcal{L}_{\text{q}} + \lambda_{\text{bal}} \mathcal{L}_{\text{bal}} + \lambda_{\text{dcr}} \mathcal{L}_{\text{dcr}} + \lambda_{\text{cons}} \mathcal{L}_{\text{cons}}. \tag{3}$$

After training, we fix $f_\theta$ and use it as a semantic hashing module within the SemBind framework.

### 3.2. Watermark Embedding

**Mask expansion and permutation.** To bind a watermarked initial latent to the semantic binary code $m = f_\theta(x) \in \{0,1\}^B$, we expand $m$ into a spatial mask aligned with the shape as the diffusion initial latent via tiling. A mask ratio $\sigma$ controls the fraction of latent coordinates being modulated: larger $\sigma$ strengthens semantic binding but may reduce robustness to benign perturbations. Since tiling introduces periodic structure, we further shuffle the mask with a secret permutation key so that the modulation is spatially dispersed and less structured.

Specifically, let the initial latent of the diffusion model be $z_T \in \mathbb{R}^{C \times H \times W}$ with $L = CHW$ coordinates. Given a semantic code $m = f_\theta(x) \in \{0,1\}^B$, a mask ratio $\sigma \in [0,1]$, and a *secret* permutation key $K_{\text{perm}}$ shared between

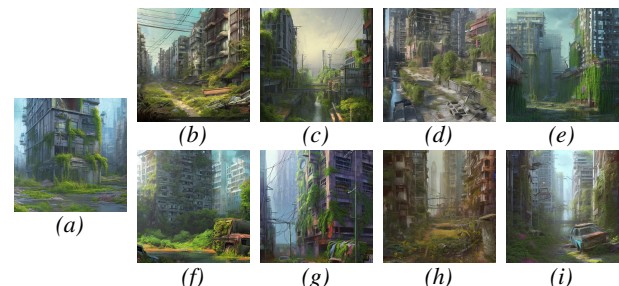

*(a)*

*(b)*    *(c)*    *(d)*    *(e)*

*(f)*    *(g)*    *(h)*    *(i)*

*Figure 4.* Visual comparison of different latent-based watermarking methods and their SemBind-enhanced variants. (a) original unwatermarked image; (b) Tree-Ring; (c) Gaussian Shading; (d) PRC; (e) Gaussian Shading++; (f) Tree-Ring (SemBind); (g) Gaussian Shading (SemBind); (h) PRC (SemBind); (i) Gaussian Shading++ (SemBind). All images are generated from the prompt: *"Post apocalyptic city overgrown abandoned city, highly detailed, art by Range Murata, highly detailed, 3d, octane render, bright colors, digital painting, trending on artstation, sharp focus."*

embedder and verifier, we construct a spatial binary mask by: (i) Tiling: form $\tilde{m} \in \{0,1\}^L$ by repeating the bits of $m$ and truncating to length $L$; (ii) Ratio selection: let $L_\sigma = \lfloor \sigma L \rfloor$ and obtain $\tilde{m}_\sigma$ by keeping the first $L_\sigma$ entries of $\tilde{m}$ and setting the rest to 0; (iii) Permutation: use $K_{\text{perm}}$ to define a fixed permutation $\pi$ over $\{1, \dots, L\}$ and set $\tilde{m}_\sigma'[i] = \tilde{m}_\sigma[\pi(i)]$ for $i = 1, \dots, L$; (iv) Reshape: reshape $\tilde{m}_\sigma'$ back to $\{0,1\}^{C \times H \times W}$.

We then map this binary mask to a bipolar *sign mask*: $S(x; \sigma, K_{\text{perm}}) = 1 - 2\, \tilde{m}_\sigma' \in \{-1, +1\}^{C \times H \times W}$.

**SemBind embedding.** Let $\mathcal{W}_{\text{embed}}$ denote a latent-based watermarking scheme that, given a message $m_w$ and watermark key $K_w$, produces a watermarked initial latent $z_T^{(w)} = \mathcal{W}_{\text{embed}}(m_w, K_w)$.

For a given prompt $p$, the service provider first randomly generates an auxiliary clean image $x$ using the same diffusion model $\Theta$. The semantic masker in *binary mode* produces a code $m = f_\theta(x)$, which is expanded into a sign mask $S(x; \sigma, K_{\text{perm}})$. We then define the *semantically bound* initial latent by element-wise multiplication: $z_T^{(w,\text{sem})} = S(x; \sigma, K_{\text{perm}}) \odot z_T^{(w)}$.

The final watermarked image is obtained by running the denoising scheduler from $z_T^{(w,\text{sem})}$ down to $z_0^{(w,\text{sem})}$ and decoding $z_0^{(w,\text{sem})}$ with the VAE decoder $\mathcal{D}$.

### 3.3. Watermark Extraction

Given a watermarked image $\hat{x}$, SemBind performs watermark extraction in three steps: we first run the semantic masker to obtain a binary code and its latent mask $\hat{S} = S(\hat{x}; \sigma, K_{\text{perm}}) \in \{-1, +1\}^{C \times H \times W}$, where $K_{\text{perm}}$ is the secret permutation key shared with the embedding

process; then the image $\hat{x}$ is passed through the VAE encoder $\mathcal{E}$ to obtain a latent representation $\hat{z}_0 = \mathcal{E}(\hat{x})$, and inverted with the inversion scheduler to estimate the (semantically bound) initial noise $\hat{z}_T^{(w,\text{sem})} = \mathcal{I}_{0 \to T}(\hat{z}_0; u)$; finally, we unbind the semantic modulation by computing $\hat{z}_T^{(w)} = \hat{S} \odot \hat{z}_T^{(w,\text{sem})}$ and feed $\hat{z}_T^{(w)}$ into the decoder of the underlying watermark scheme to obtain the extracted watermark $\hat{m}_w = \mathcal{W}_{\text{decode}}\left( \hat{z}_T^{(w)} \right)$.

# 4. Experiments

## 4.1. Experimental Setup

In our primary experiments, we focus on text-to-image latent diffusion models and adopt *Stable Diffusion v2.1*[1]. All images are generated at a resolution of $512 \times 512$ with a latent space of $4 \times 64 \times 64$. During sampling, we use classifier-free guidance with a scale of 7.5 and run 50 denoising steps using DPMSolver (Lu et al., 2022). For watermark extraction and forgery attack, we perform diffusion inversion scheduler using the exact inversion method of Hong *et al.* (Hong et al., 2024) to obtain the latent $z_T$. Unless otherwise specified, we use 50 inversion steps and an inverse order of 0.

**Training details.** We instantiate the semantic image encoder backbone as DINOv2-Giant (Oquab et al., 2023) and train the Semantic Masker $f_\theta$ on our large prompt-conditioned DINO embedding corpus (SemCon-3M, in Appendix A). Full training details are provided in Appendix B.

**Watermarking methods and datasets for evaluation.** We evaluate SemBind on four representative latent-based watermarking schemes: Tree-Ring (Wen et al., 2023), Gaussian Shading (Yang et al., 2024b), PRC watermark (Gunn et al.), and Gaussian Shading++ (Yang et al., 2025). For each scheme, we compare the original baseline with its SemBind-enhanced variant (denoted "-S"). Unless stated otherwise, we follow the default settings of the respective papers for both embedding and detection. For detection, we set the decision threshold to achieve a (theoretical) false positive rate (FPR) of $10^{-6}$ and report the corresponding true positive rate (TPR). For SemBind, we set the mask ratio to $\sigma = 1$ for Tree-Ring and Gaussian Shading, and $\sigma = 0.5$ for PRC and Gaussian Shading++.

All the evaluations are performed on two prompt sets: the MS-COCO (Lin et al., 2014) and the Stable Diffusion Prompts (SDP) dataset[2]. All experiments are conducted on NVIDIA RTX A6000 GPUs.

[1]https://huggingface.co/stabilityai/stable-diffusion-2-1-base
[2]https://huggingface.co/datasets/Gustavosta/Stable-Diffusion-Prompts

*Table 1.* Quality of watermarked images.

| Method | FID↓ | | CLIP score↑ | |
|---|---|---|---|---|
| | value | $t$-value | value | $t$-value |
| SD 2.1 | $25.23 \pm .18$ | – | $0.3629 \pm .0006$ | – |
| TR | $25.43 \pm .13$ | 2.581 | $0.3632 \pm .0006$ | 0.8278 |
| TR-S | $25.27 \pm .20$ | 0.5229 | $0.3633 \pm .0010$ | 0.9526 |
| GS | $25.20 \pm .22$ | 0.3567 | $0.3631 \pm .0005$ | 0.6870 |
| GS-S | $25.28 \pm .20$ | 0.5714 | $0.3634 \pm .0011$ | 1.055 |
| PRC | $25.23 \pm .14$ | 0.1772 | $0.3624 \pm .0006$ | 0.8176 |
| PRC-S | $25.24 \pm .10$ | 0.2412 | $0.3625 \pm .0006$ | 1.136 |
| GS++ | $25.22 \pm .10$ | 0.1597 | $0.3626 \pm .0011$ | 0.6228 |
| GS++-S | $25.23 \pm .12$ | 0.3855 | $0.3627 \pm .0008$ | 0.8646 |

## 4.2. Quality Comparison

We assess watermarked image quality using FID (Heusel et al., 2017) and CLIP score (Radford et al., 2021). Following Tree-Ring (Wen et al., 2023), we compute FID using 5,000 MS-COCO prompts and paired real images as ground truth. For CLIP, we use 1,000 prompts from the SDP dataset and their generated images. For each method, we run 10 trials and report the mean and variance.

To further verify that our method maintains the provable undetectability property of the underlying method, we also perform a $t$-test using the ten sets of data from each method. Specifically, the goal is to confirm that the metrics of the watermarking method and those of SD 2.1 are statistically indistinguishable, where a smaller $t$-value indicates better results.

The experimental results are shown in Table 1. For the non-provable undetectability method Tree-Ring (Wen et al., 2023), the addition of SemBind actually *improves* FID performance. This is because the semantic mask is generated independently of the Tree-Ring perturbation, so their multiplicative interaction acts as a mild, semantics–conditioned randomization of the initial latent, enriching its distribution and slightly improving sample quality.

For provable undetectability methods (Yang et al., 2024b; Gunn et al.; Yang et al., 2025), the small fluctuations in the $t$-values indicate that SemBind does not introduce any measurable degradation in FID or CLIP, and are fully consistent with the fact that these schemes remain provable undetectable under semantic masking.

We theoretically prove that SemBind preserves the *provable undetectability* of the underlying latent-based watermark, as formalized in the following theorem.

**Theorem 4.1** (**Semantic masking preserves provable undetectability (*informal*)**). *For latent-based watermarking scheme $\mathcal{W}$ that is provably undetectable in the single-sample (resp. multi-sample) setting, its SemBind variant $\mathcal{W}^{\text{sem}}$ remains provably undetectable in the same setting.*

*Table 2.* Imprint forgery attack on the SDP dataset for four latent-based watermarking schemes and their SemBind-enhanced variants.

| Method | Step | Attacker Model: SD 2.1 | | | Attacker Model: SD 1.5 | | |
|---|---|---|---|---|---|---|---|
| | | Det.↓ | Bit Acc.↓ | PSNR | Det.↓ | Bit Acc.↓ | PSNR |
| TR | 50/100/150 | 1.00/1.00/1.00 | — | 23.32/22.12/21.34 | 1.00/1.00/1.00 | — | 22.88/21.85/21.17 |
| TR-S | 50/100/150 | 0.01/0.02/0.02 | — | 23.33/22.14/21.38 | 0.01/0.01/0.01 | — | 22.88/21.83/21.16 |
| GS | 50/100/150 | 1.00/1.00/1.00 | 0.9998/1.0000/1.0000 | 23.36/22.16/21.40 | 1.00/1.00/1.00 | 0.9993/0.9998/0.9998 | 22.89/21.86/21.19 |
| GS-S | 50/100/150 | 0.04/0.06/0.07 | 0.4818/0.4843/0.4925 | 23.34/22.14/21.37 | 0.03/0.04/0.05 | 0.4893/0.4918/0.4858 | 22.90/21.85/21.17 |
| PRC | 50/100/150 | 1.00/1.00/1.00 | 1.0000/1.0000/1.0000 | 23.37/22.17/21.40 | 0.99/1.00/1.00 | 0.9949/1.0000/1.0000 | 22.93/21.89/21.22 |
| PRC-S | 50/100/150 | 0.06/0.18/0.25 | 0.5266/0.5559/0.5064 | 23.39/22.19/21.42 | 0.00/0.01/0.02 | 0.5015/0.5067/0.5067 | 22.93/21.90/21.21 |
| GS++ | 50/100/150 | 1.00/1.00/1.00 | 0.9995/0.9999/1.0000 | 23.37/22.18/21.42 | 0.98/0.99/1.00 | 0.9803/0.9881/0.9944 | 22.91/21.88/21.19 |
| GS++-S | 50/100/150 | 0.15/0.36/0.49 | 0.5699/0.6720/0.7291 | 23.37/22.17/21.41 | 0.02/0.05/0.09 | 0.5063/0.5222/0.5383 | 22.91/21.88/21.20 |

*Table 3.* Imprint forgery attack on the COCO dataset for four latent-based watermarking schemes and their SemBind-enhanced variants.

| Method | Step | Attacker Model: SD 2.1 | | | Attacker Model: SD 1.5 | | |
|---|---|---|---|---|---|---|---|
| | | Det.↓ | Bit Acc.↓ | PSNR | Det.↓ | Bit Acc.↓ | PSNR |
| TR | 50/100/150 | 0.99/1.00/1.00 | — | 23.34/22.15/21.38 | 0.99/1.00/1.00 | — | 22.88/21.84/21.15 |
| TR-S | 50/100/150 | 0.08/0.08/0.08 | — | 23.33/22.13/21.36 | 0.07/0.07/0.07 | — | 22.86/21.82/21.14 |
| GS | 50/100/150 | 1.00/1.00/1.00 | 0.9998/0.9999/1.0000 | 23.34/22.15/21.37 | 1.00/1.00/1.00 | 0.9960/1.0000/1.0000 | 22.87/21.83/21.15 |
| GS-S | 50/100/150 | 0.09/0.10/0.10 | 0.5084/0.5060/0.5077 | 23.32/22.11/21.34 | 0.08/0.09/0.10 | 0.5078/0.5122/0.5122 | 22.87/21.82/21.13 |
| PRC | 50/100/150 | 1.00/1.00/1.00 | 1.0000/1.0000/1.0000 | 23.37/22.15/21.38 | 0.99/1.00/1.00 | 0.9941/1.0000/1.0000 | 22.92/21.87/21.18 |
| PRC-S | 50/100/150 | 0.08/0.14/0.27 | 0.5444/0.5547/0.6211 | 23.37/22.17/21.40 | 0.05/0.05/0.06 | 0.5241/0.5241/0.5295 | 22.92/21.89/21.19 |
| GS++ | 50/100/150 | 1.00/1.00/1.00 | 0.9994/0.9999/0.9999 | 23.36/22.15/21.37 | 0.99/1.00/1.00 | 0.9845/0.9951/0.9964 | 22.92/21.87/21.17 |
| GS++-S | 50/100/150 | 0.15/0.43/0.59 | 0.5050/0.7096/0.7769 | 23.36/22.16/21.39 | 0.05/0.07/0.13 | 0.5166/0.5323/0.5586 | 22.92/21.88/21.19 |

*Table 4.* Reprompt forgery attack for four latent-based watermarking schemes and their SemBind-enhanced variants.

| Method | SD 2.1 attacker | | | | SD 1.5 attacker | | | |
|---|---|---|---|---|---|---|---|---|
| | SDP | | COCO | | SDP | | COCO | |
| | Det.↓ | Bit Acc.↓ | Det.↓ | Bit Acc.↓ | Det.↓ | Bit Acc.↓ | Det.↓ | Bit Acc.↓ |
| TR | 0.99 | — | 1.00 | — | 1.00 | — | 1.00 | — |
| TR-S | 0.54 | — | 0.14 | — | 0.48 | — | 0.18 | — |
| GS | 1.00 | 0.9887 | 0.99 | 0.9788 | 1.00 | 0.9895 | 1.00 | 0.9823 |
| GS-S | 0.60 | 0.7054 | 0.06 | 0.4894 | 0.56 | 0.6867 | 0.10 | 0.5025 |
| PRC | 0.95 | 0.9745 | 0.93 | 0.9662 | 0.94 | 0.9711 | 0.93 | 0.9656 |
| PRC-S | 0.51 | 0.7338 | 0.50 | 0.7287 | 0.12 | 0.5622 | 0.11 | 0.5571 |
| GS++ | 0.91 | 0.9527 | 0.84 | 0.9132 | 0.69 | 0.8377 | 0.57 | 0.7732 |
| GS++-S | 0.43 | 0.7044 | 0.25 | 0.6150 | 0.15 | 0.5679 | 0.03 | 0.5107 |

Intuitively, these schemes produce an initial watermarked latent that is (computationally) indistinguishable from a standard Gaussian, and SemBind only multiplies this latent by an independent $\{\pm 1\}$ sign mask, which leaves the Gaussian distribution invariant. A formal statement of the security game and the full proof are provided in Appendix H.

### 4.3. Defense Against Forgery Attack

We evaluate SemBind under the two canonical black-box forgery attacks introduced by Müller et al. (Müller et al., 2025): the *imprinting attack* and the *reprompting attack*.

**Imprinting attack.** We consider two attacker models: *Stable Diffusion v2.1* (the "match" case) and *Stable Diffusion v1.5* (the "mismatch" case). For each watermarking scheme and attacker model, we generate 100 watermarked images on both SDP and COCO prompt validation sets. As target

images for the attacker, we randomly sample 100 natural photographs from the COCO natural image validation set. Following (Müller et al., 2025), the attacker optimizes the target latents for 150 gradient steps with a learning rate of 0.01, and we probe watermark detection every 50 steps.

Table 2 and Table 3 report the imprinting results. When the attacker uses SD 2.1 (the "match" case), SemBind already results in substantial gains: for Tree-Ring and Gaussian Shading, it reduces the detection rate (Det.) from essentially 100% to below 10% on both SDP and COCO, while the bit accuracy for GS decreases from nearly perfect ($\approx 1.0$) to around 0.5, i.e., close to random guessing. For PRC and Gaussian Shading++, SemBind again cuts Det. by more than half and lowers Bit Acc. by over 20%.

When the attacker instead uses SD 1.5 (the "mismatch" case), SemBind becomes even more effective: across all four watermarking schemes, Det. is driven to very low values and Bit Acc. is pushed much closer to 0.5, corresponding to near-random decoding. This indicates that SemBind's defensive advantage increases as the attacker's model becomes more mismatched to the defended generator, yielding near-perfect protection against imprinting in this setting. This scenario is also closer to practical real-world deployments, where attackers typically have access only to the mismatched diffusion models rather than the exact one.

**Reprompting attack.** For each of the four latent-based watermarking schemes and their SemBind-enhanced variants, we use the same SDP and COCO *prompt-evaluation*

*Table 5.* Robustness test on SDP. "Average (Distortion)" represents the mean value across all distortion types.

| Method | None (Det./Acc.) | JPEG (QF=70) | Brightness | GauBlur ($r$=3) | GauNoise ($\sigma$=0.01) | MedFilter ($k$=7) | Resize (×0.5) | Average (Distortion) |
|---|---|---|---|---|---|---|---|---|
| TR | 1.00 | 0.98 | 1.00 | 1.00 | 1.00 | 1.00 | 1.00 | 0.997 |
| TR-S | 1.00 | 0.98 | 1.00 | 0.99 | 0.99 | 0.99 | 1.00 | 0.992 |
| GS | 1.00/1.0000 | 0.99/0.9996 | 1.00/0.9966 | 1.00/0.9966 | 1.00/0.9989 | 1.00/0.9984 | 1.00/0.9999 | 0.998/0.99833 |
| GS-S | 1.00/0.9982 | 0.99/0.9871 | 0.99/0.9888 | 0.99/0.9796 | 0.99/0.9894 | 1.00/0.9884 | 1.00/0.9958 | 0.993/0.98818 |
| PRC | 1.00/1.0000 | 1.00/1.0000 | 1.00/1.0000 | 1.00/1.0000 | 1.00/1.0000 | 0.97/0.9807 | 1.00/1.0000 | 0.995/0.99678 |
| PRC-S | 1.00/1.0000 | 0.99/0.9902 | 0.98/0.9850 | 0.86/0.9271 | 0.98/0.9896 | 0.83/0.9250 | 1.00/1.0000 | 0.940/0.96948 |
| GS++ | 1.00/1.0000 | 1.00/0.9973 | 0.97/0.9841 | 0.98/0.9765 | 0.93/0.9599 | 0.96/0.9707 | 1.00/0.9988 | 0.973/0.98122 |
| GS++-S | 1.00/0.9998 | 0.96/0.9752 | 0.96/0.9785 | 0.94/0.9466 | 0.89/0.9402 | 0.85/0.9138 | 0.99/0.9932 | 0.932/0.95792 |

*Table 6.* Robustness test on COCO. "Average (Distortion)" represents the mean value across all distortion types.

| Method | None (Det./Acc.) | JPEG (QF=70) | Brightness | GauBlur ($r$=3) | GauNoise ($\sigma$=0.01) | MedFilter ($k$=7) | Resize (×0.5) | Average (Distortion) |
|---|---|---|---|---|---|---|---|---|
| TR | 1.00 | 1.00 | 1.00 | 1.00 | 1.00 | 1.00 | 1.00 | 1.000 |
| TR-S | 1.00 | 1.00 | 0.99 | 1.00 | 1.00 | 1.00 | 1.00 | 0.998 |
| GS | 1.00/1.0000 | 1.00/0.9980 | 1.00/0.9984 | 1.00/0.9967 | 1.00/0.9988 | 1.00/0.9980 | 1.00/1.0000 | 1.000/0.99832 |
| GS-S | 1.00/0.9996 | 1.00/0.9934 | 1.00/0.9915 | 1.00/0.9830 | 0.99/0.9896 | 1.00/0.9871 | 1.00/0.9964 | 0.998/0.99017 |
| PRC | 1.00/1.0000 | 0.99/0.9948 | 0.95/0.9751 | 0.96/0.9854 | 1.00/1.0000 | 0.91/0.9643 | 1.00/1.0000 | 0.968/0.98660 |
| PRC-S | 1.00/1.0000 | 0.97/0.9855 | 0.91/0.9593 | 0.84/0.9261 | 0.99/0.9945 | 0.80/0.8910 | 1.00/1.0000 | 0.918/0.95940 |
| GS++ | 1.00/1.0000 | 0.97/0.9796 | 0.98/0.9877 | 0.97/0.9700 | 0.87/0.9259 | 0.95/0.9598 | 1.00/0.9980 | 0.957/0.97017 |
| GS++-S | 1.00/0.9998 | 0.96/0.9752 | 0.96/0.9757 | 0.92/0.9388 | 0.84/0.9146 | 0.89/0.9279 | 0.98/0.9864 | 0.925/0.95310 |

sets, each containing 100 prompts. We use mismatched text prompts sampled from the Inappropriate Image Prompts (I2P) dataset[3], with 100 mismatched prompts in total, for attack. As attacker models, we again consider both *Stable Diffusion v2.1* and *Stable Diffusion v1.5*.

Table 4 summarizes the reprompting results. Across all watermarking schemes, SemBind consistently and substantially attenuates forgery: for every setting, the SemBind variants reduce the detection rate (Det.) by more than $40\%$ and lower bit accuracy (Bit Acc.) by at least $25\%$ compared to the original watermark baselines. The gains are particularly pronounced on COCO, where Det. often decreases to $20\%$ or less and Bit Acc. approaches $0.5$, corresponding to near-random message recovery. As in the imprinting case, SemBind is even more effective against the mismatch-case attacker (SD 1.5), where forged images are rarely accepted and recovered payloads are close to random.

### 4.4. Robustness

To evaluate the robustness of the method, we selected several common image manipulations: (a) JPEG Compression, $QF = 70$ (JPEG). (b) Brightness, $factor = 1$. (c) Gaussian Blur, $radius = 3$ (GauBlur). (d) Gaussian Noise, $\mu = 0$, $\sigma = 0.01$ (GauNoise). (e) Median Filtering, $kernel\_size = 7$ (MedFilter). (f) 50% Resize and restore (Resize). We conducted experiments on the MS-COCO and SDP datasets, respectively, testing 100 watermarked images under each type of distortion. For each method, we report the metrics under each distortion type and compute the average across all distortions. The experimental results are

---

[3] https://huggingface.co/datasets/AIML-TUDA/i2p

presented in Table 5 and Table 6.

Overall, SemBind causes only a mild robustness drop across schemes. The impact on Tree-Ring and Gaussian Shading is particularly small, and their detection performance remains close to the original baselines under all tested distortions. For PRC and Gaussian Shading++, we observe a moderate but still acceptable degradation: the true positive rate decreases by about 3–5% on average and bit accuracy drops by about 2–3%, with the most visible gaps under Gaussian blur and median filtering. This trend is consistent with the underlying robustness of the base watermarks: schemes that are robust to common perturbations remain robust after adding SemBind, while schemes with weaker baseline robustness exhibit larger drops in their SemBind-enabled variants.

## 5. Limitations and Discussion

SemBind introduces additional training cost to learn the semantic masker. In typical platform settings, however, this is a one-time investment: since it interfaces with the system only through a semantic code and a masking operation in the initial latent, upgrading the underlying latent watermark scheme does not require retraining the masker.

SemBind requires generating an auxiliary clean image at deployment time to extract the semantic code used for masking. This additional generation step introduces more than a single standard generation pass, and therefore increases the computational cost compared with directly applying a latent watermark. Moreover, because SemBind relies on two generations, a potential failure case arises when the auxiliary clean image and the final watermarked image are semantically inconsistent. In such cases, the semantic code extracted from the auxiliary image may not properly match

the semantics of the watermarked image, weakening the intended semantic binding. A simple practical mitigation is to perform a self-check after generation and re-run SemBind with a newly generated auxiliary image when the consistency check fails. However, this mitigation further increases the computational overhead. This overhead is necessary to preserve provable undetectability: maintaining undetectability typically requires embedding to occur entirely in the initial latent space, yet at that point, no image is available to infer semantics from. A possible solution is to start from the prompt, but this faces the challenge of aligning the prompt with the generated image.

SemBind assumes that the trained semantic masker is kept private by the service provider. If an attacker obtains the exact deployed masker and can directly execute the bind/unbind operations, the anti-forgery mechanism may no longer be effective; such a fully white-box attacker is outside our threat model. A promising future direction is to protect the masker via server-side verification or introduce key-dependent semantic masking.

Moreover, we defer further discussion of additional black-box forgery attacks to Appendix G, adaptive attackers to Appendix E, additional generalization experiments to Appendix F, and the evaluation under watermark removal attacks to Appendix D. Our experiments suggest that SemBind generalizes well across different models and prompt distributions, and remains effective against additional black-box forgery attacks such as **WMCopier** (Dong et al., 2025). They also show that adaptive attackers cannot easily train a surrogate semantic masker or reliably spoof it in the pixel domain, and that the semantic masking mechanism does not noticeably amplify the effectiveness of watermark removal attacks such as **CtrlRegen** (Liu et al., 2025).

## 6. Conclusion

We propose *SemBind*, the first defense for latent-based diffusion watermarks against black-box forgery by binding latent watermark signals to image semantics via a learned semantic masker. SemBind applies broadly to existing latent-based schemes. It reduces false acceptance under imprinting and reprompting, and preserves provable undetectability whenever the underlying scheme has it.

## Acknowledgements

This work was supported by the National Natural Science Foundation of China (Grants U2436601, U2336206, and 62472398) and by the New Generation Artificial Intelligence–National Science and Technology Major Project (No. 2025ZD0123202).

## Impact Statement

This paper studies defenses for latent-based watermarking in diffusion models, with the goal of improving provenance tracking and copyright authentication in the presence of black-box forgery attacks. If deployed responsibly, our method can help AI service providers and content platforms better identify and deter the unauthorized generation of misleading or harmful content by preventing attackers from transferring a valid watermark to illicit semantics. We believe the primary impact of this work is positive: it advances the reliability of watermark-based provenance for generative models while making explicit the assumptions and limitations under which such defenses are effective.

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

# A. Dataset Construction (SemCon-3M)

We next describe how **SemCon-3M** is built. Constrained by computational resources, we first mine and cluster prompts from the MS-COCO-2017 (Lin et al., 2014) and the Stable Diffusion Prompts (SDP) dataset[4] to form a compact and representative prompt set. To train a semantic masker that yields similar outputs for images generated from the same prompt (and their semantics-preserving transforms), we then generate multiple images per prompt with *Stable Diffusion v2.1*[5] and apply semantics-preserving augmentations. Finally, we summarize the resulting dataset composition and discuss how the COCO/SDP imbalance influences downstream results.

**Prompt Mining and Clustering.** We collect caption prompts from the training split of MS-COCO-2017 (Lin et al., 2014) and from the `train.parquet` file of the Stable Diffusion Prompts (SDP) dataset[6], normalize whitespace, and deduplicate at the string level. Each prompt is embedded with a sentence-transformer *BGE-large-en-v1.5*[7] and $\ell_2$-normalized so that Euclidean distance rankings coincide with cosine similarity, making $k$-means effectively operate in the spherical setting. To scale to millions of prompts, we run Mini-Batch $k$-means (Sculley, 2010) with $K = 64{,}000$ clusters. Within each cluster we rank candidates by proximity to the centroid and select $M_i \in \{1, 2\}$ prompts in proportion to the cluster's size, enforcing intra-cluster diversity by rejecting any candidate whose cosine similarity to already selected ones exceeds $0.8$. We do this to preferentially select prompts that are semantically typical. This produces $\approx 96{,}041$ representative prompts that cover the prompt distribution while avoiding near duplicates.

**Data Generation and Augmentation.** For each selected prompt, we synthesize **16** independent images with *Stable Diffusion v2.1* (guidance scale $= 7.5$, 50 denoising steps with DPMSolver (Lu et al., 2022), output $512\times512$, latent $4\times64\times64$). We then create **16** additional, semantics–preserving views by applying one augmentation per original, sampled from a fixed operator pool and executed in a stable order (composition $\rightarrow$ geometry $\rightarrow$ resolution/compression $\rightarrow$ color/blur/noise $\rightarrow$ flip), which avoids black borders and preserves content semantics. The pool includes:

- **RandomResizedCrop** (scale $0.4-1.0$, aspect $3{:}4-4{:}3$, antialias).

- **RandomPerspective** (distortion_scale $= 0.25$).

- **RandomAffine** (degrees $\pm15°$, translate $\leq 10\%$, scale $0.9-1.1$, shear $\pm(5°, 3°)$, bicubic).

- **ResizeDownUp** (downscale ratio $0.5-0.85$, then upscale back).

- **JPEG Compression** (quality $55-85$).

- **ColorJitter** (brightness/contrast/saturation $0.3$).

- **RandomGrayscale**.

- **Gaussian Blur** (Gaussian radius $0.8-1.5$).

- **Additive Gaussian noise** (std $0.005-0.03$).

- **Light salt&pepper noise** (prob $0.002-0.01$).

- **Deterministic horizontal flip**.

For each generated image, we randomly select one or two operators from the above pool and apply them to that image to obtain one augmented view. This yields 32 views per prompt (16 generations + 16 augmented).

---

[4] https://huggingface.co/datasets/Gustavosta/Stable-Diffusion-Prompts
[5] https://huggingface.co/stabilityai/stable-diffusion-2-1-base
[6] https://huggingface.co/datasets/Gustavosta/Stable-Diffusion-Prompts
[7] https://huggingface.co/BAAI/bge-large-en-v1.5

**Final Dataset Composition.** From the curated pool of 96,041 prompts (selected from COCO and SDP), we generate 32 views per prompt (16 independent generations and 16 semantics-preserving augmentations). We then extract DINOv2-giant (Oquab et al., 2023) CLS embeddings (dimension 1536) for all images, yielding **SemCon-3M**: a corpus of $96{,}041 \times 32 = 3{,}073{,}312$ embeddings. By source, COCO contributes 84,675 prompts (88.2%; 2,709,600 embeddings) and SDP contributes 11,366 prompts (11.8%; 363,712 embeddings), as summarized in Table 7. Because COCO dominates the training distribution, we consistently observe stronger performance on COCO than on SDP, which we attribute to this dataset imbalance rather than a method-specific bias.

*Table 7.* SemCon-3M composition by source. Each prompt contributes 32 views (images/embeddings).

| Source | # Embeddings | % Embeddings |
|--------|--------------|--------------|
| COCO   | **2,709,600** | **88.2%** |
| SDP    | **363,712**   | **11.8%** |
| Total  | **3,073,312** | 100% |

Because COCO comprises the vast majority of **SemCon-3M**, the semantic masker is trained more fully on COCO-style prompts and imagery. As a result, we consistently observe stronger anti-forgery resistance (lower false-accept rates under imprinting/reprompting) and higher robustness to common perturbations on COCO than on SDP. We attribute this gap to data imbalance rather than a method-specific bias. The experimental result is analyzed in Section C.

## B. Model Architecture & Training Details

**Backbone & Embeddings.** We adopt a frozen DINOv2-giant (Oquab et al., 2023) vision encoder to obtain a single global CLS embedding $e \in \mathbb{R}^{1536}$ per image. All images are preprocessed using the standard DINOv2 pipeline (resize/crop and normalization to the encoder's default statistics). Unless otherwise stated, CLS embeddings are $\ell_2$–normalized before being fed to our hashing network.

**Network Architecture Details.** The semantic masker $f_\theta$ is a lightweight MLP-style hashing network with three modules:

- **Encoder** $\mathrm{Enc} : \mathbb{R}^{1536} \to \mathbb{R}^H$: a stack of residual fully-connected blocks (depth 2 by default) with BatchNorm and GELU; we set hidden width $H = 2048$.

- **Projection head** $\mathrm{Proj} : \mathbb{R}^H \to \mathbb{R}^D$: a small MLP that maps to $D$-dimensional features, followed by $\ell_2$ normalization; we use $D = 8192$ in experiments.

- **Hash head** $\mathrm{Hash} : \mathbb{R}^D \to \mathbb{R}^B$: residual fully-connected blocks ending in a linear layer that outputs $B$ hash logits; we use $B = 1024$.

At inference, given CLS embedding $e$, the network first produces a *hash logit* $\ell = \mathrm{Hash}(\mathrm{Proj}(\mathrm{Enc}(e))) \in \mathbb{R}^B$, then a *soft binary code* $b = \tanh(s\,\ell) \in [-1, 1]^B$, and finally a *binary code* $m = (\mathrm{sign}(b) + 1)/2 \in \{0, 1\}^B$, where $s > 0$ controls the soft sign's sharpness.

**Training Schedule: losses, temperatures, and hardness.** We train $f_\theta$ in two stages, operating in *logit mode* (i.e., losses consume logits or their $\tanh$ transforms).

**Stage-1 (feature contrast).** We optimize $\mathrm{Enc}+\mathrm{Proj}$ with supervised contrastive loss on normalized features $z_i \in \mathbb{R}^D$:

$$\mathcal{L}_{\mathrm{sup}} = -\frac{1}{N} \sum_{i=1}^{N} \frac{1}{|P(i)|} \sum_{p \in P(i)} \log \frac{\exp(z_i^\top z_p / \tau)}{\sum_{a \neq i} \exp(z_i^\top z_a / \tau)}.$$

We use Adam with $\mathrm{lr} = 1 \times 10^{-4}$, gradient clipping 1.0, temperature $\tau = 0.07$ for the first half of training and 0.10 for the second half. Default epochs: `epoch1` = 180. **Stage-2 (hash contrast + regularizers).** We freeze the backbone and Stage-1 modules, and train $\mathrm{Hash}$ to produce near-binary, balanced, decorrelated codes. Let $\ell_i$ be logits and $b_i = \tanh(s\,\ell_i)$ the soft

binary codes. We apply a supervised contrastive loss in code space with temperature $\tau_h$,

$$\mathcal{L}_{\text{hash}} = -\frac{1}{N} \sum_{i=1}^{N} \frac{1}{|P(i)|} \sum_{p \in P(i)} \log \frac{\exp\big((b_i^\top b_p/B)/\tau_h\big)}{\sum_{a \neq i} \exp\big((b_i^\top b_a/B)/\tau_h\big)},$$

and add three standard regularizers on $\{b_i\}$: quantization $\mathcal{L}_{\text{q}} = \mathbb{E}[1 - |b_i|]$, bit-balance $\mathcal{L}_{\text{bal}} = \frac{1}{B}\sum_{k=1}^{B}\big(\frac{1}{N}\sum_i b_{i,k}\big)^2$, decorrelation $\mathcal{L}_{\text{dcr}} = \|C - I\|_F^2$ (where $C$ is the batch covariance). To stabilize codes, we jitter features twice and penalize their discrepancy $\mathcal{L}_{\text{cons}} = \mathbb{E}[\|b_i^{(1)} - b_i^{(2)}\|_1]$. The stage objective is

$$\mathcal{L} = \mathcal{L}_{\text{hash}} + \lambda_{\text{q}}\mathcal{L}_{\text{q}} + \lambda_{\text{bal}}\mathcal{L}_{\text{bal}} + \lambda_{\text{dcr}}\mathcal{L}_{\text{dcr}} + \lambda_{\text{cons}}\mathcal{L}_{\text{cons}}.$$

We use Adam with $\text{lr} = 1 \times 10^{-4}$, jitter $\sigma = 0.01$ (no feature dropout), and a *hardness* schedule for $s$: $s\_\text{init} = 1.0$ and at epochs $\{20, 40, 70\}$ we multiply $s$ by $\gamma = 2.5$. Default epochs: `epoch2` $= 160$. (At evaluation, we use a fixed inference sharpness, e.g., $s = 12$ to binarize.)

*Note on ablations.* The effect of `epoch1`/`epoch2` is analyzed in Sec. C; here we keep the schedule fixed and report the default settings above.

Tables 8 and 9 summarize the model architecture and training hyperparameters; we will publicly release our code to the research community.

*Table 8.* Training objectives and hyperparameters.

| Setting | Value |
|---|---|
| Widths $(H, D, B)$ | 2048, 8192, 1024 |
| Norm/Act | BN (eps $10^{-5}$, mom 0.1), GELU |
| Stage-1 loss | SupCon; $\tau$ : $0.07 \rightarrow 0.10$ (mid-training) |
| Stage-1 opt | Adam, lr $1 \times 10^{-4}$, grad-clip 1.0 |
| Stage-1 epochs | 180 |
| Stage-2 loss | Hash SupCon ($\tau_h = 0.10$) |
| Regularizers | $\lambda_q = 0.10$, $\lambda_{\text{bal}} = 0.10$, |
| | $\lambda_{\text{dcr}} = 0.10$, $\lambda_{\text{cons}} = 0.20$ |
| Hardness $s$ | $1.0 \xrightarrow{\times 2.5}$ at epochs $20/40/70$ |
| Stage-2 opt | Adam, lr $1 \times 10^{-4}$; jitter $\sigma = 0.01$ |
| Stage-2 epochs | 160 |
| PK sampling | $P = 64$, $K = 16$ (batch $= PK$) |
| Inference $s$ | 12 (for `tanh` at test) |

*Table 9.* Semantic masker architecture. DINOv2-giant is frozen and we use its CLS embedding as input.

| Module | Layer (settings) | Output |
|---|---|---|
| Backbone | DINOv2-giant (ViT-G/14), CLS only; | $\mathbb{R}^{1536}$ |
| Enc | Linear $1536 \rightarrow 2048$ + BN + GELU | $\mathbb{R}^{2048}$ |
| | Residual FC: [Linear $2048 \rightarrow 2048$ + BN + GELU] $\times 1$ | $\mathbb{R}^{2048}$ |
| Proj | Linear $2048 \rightarrow 8192$ + BN + GELU | $\mathbb{R}^{8192}$ |
| | $\ell_2$-normalize feature $z$ | $\mathbb{R}^{8192}$ |
| Hash | Residual FC: [Linear $D \rightarrow D$ + BN + GELU] $\times 2$ | $\mathbb{R}^{D}$ |
| | Linear $D \rightarrow B$ (hash logits $\ell$) | $\mathbb{R}^{B}$ |

**Batch Construction via PK Sampling.** Our dataset contains $\sim 10^5$ prompt classes; uniform random batching would yield almost-all-negative batches, weakening the supervised contrast. We therefore adopt a PK-sampling strategy per mini-batch: sample $P$ distinct prompts and $K$ views per prompt to form a batch of size $P \times K$ with many in-class positives. Concretely,

we maintain per-class index lists, reshuffle them each epoch, and draw contiguous $K$-sized chunks per selected label; if any class runs out of $K$ samples, we finish the epoch to avoid label imbalance in incomplete batches. Unless specified, we use $P=64$, $K=16$ (batch size 1024), which balanced GPU throughput and contrastive signal quality in our setup.

## C. Experiments on Semantic Masker and Ablation Study

**Mask ratio parameter.** We further study the effect of the mask ratio parameter, which controls the fraction of latent positions modulated by the semantic mask. Intuitively, a larger mask ratio strengthens the semantic binding and improves anti-forgery capability, but may reduce robustness to benign perturbations. Conversely, a smaller mask ratio preserves more of the original latent watermark structure and improves robustness, but weakens the semantic mismatch effect under forgery attacks.

We evaluate this trade-off on SDP by varying the mask ratio in $\{1.0, 0.75, 0.5, 0.25\}$. For robustness, we report the average detection rate and bit accuracy over seven settings: no attack, JPEG compression with quality factor 70, brightness adjustment, Gaussian blur, Gaussian noise, median filtering, and resizing. For anti-forgery, we evaluate imprinting forgery using Stable Diffusion 2.1 Base as the proxy model and report the detection rate and bit accuracy at step 150.

*Table 10.* Robustness under different mask ratios on SDP. We report average Det. / Acc. over no attack, JPEG, brightness, Gaussian blur, Gaussian noise, median filtering, and resizing.

| Method | 1.0 | 0.75 | 0.5 | 0.25 |
|---|---|---|---|---|
| TR-S | 0.992 / – | 1.000 / – | 1.000 / – | 1.000 / – |
| GS-S | 0.993 / 0.988 | 1.000 / 0.995 | 1.000 / 0.998 | 1.000 / 1.000 |
| PRC-S | 0.752 / 0.882 | 0.884 / 0.938 | 0.940 / 0.969 | 0.982 / 0.990 |
| GS++-S | 0.786 / 0.896 | 0.887 / 0.942 | 0.932 / 0.958 | 0.965 / 0.984 |

*Table 11.* Imprinting forgery results under different mask ratios on SDP. We use Stable Diffusion 2.1 Base as the proxy model and report Det. / Acc. at step 150. Lower values indicate stronger resistance to forgery.

| Method | 1.0 | 0.75 | 0.5 | 0.25 |
|---|---|---|---|---|
| TR-S | 0.02 / – | 0.16 / – | 0.79 / – | 0.98 / – |
| GS-S | 0.07 / 0.493 | 0.27 / 0.764 | 0.92 / 0.952 | 1.00 / 0.988 |
| PRC-S | 0.02 / 0.498 | 0.08 / 0.541 | 0.25 / 0.506 | 0.54 / 0.781 |
| GS++-S | 0.03 / 0.516 | 0.10 / 0.608 | 0.49 / 0.729 | 0.58 / 0.802 |

Tables 10 and 11 show a clear robustness–security trade-off. Increasing the mask ratio generally improves resistance to imprinting forgery, as the forged image suffers from stronger semantic unbinding errors. However, for PRC-S and GS++-S, using a very large ratio also reduces robustness under benign perturbations. Based on this trade-off, we use ratio $= 1.0$ for TR-S and GS-S, and ratio $= 0.5$ for PRC-S and GS++-S in our main experiments.

**Semantic masker invariance and separability.** We assess (i) within-prompt tightness and (ii) across-prompt separability. For each of 1,000 random prompts (COCO or SDP), we sample $N_o=10$ originals with SD-2.1 ($512\times512$, 50 steps, guidance 7.5), create $N_d=10$ semantics-preserving distortions of the first original image, and compute $B=1024$-bit codes (inference $s=12$). Let $\text{Ham}(\mathbf{u}, \mathbf{v}) = \|\mathbf{u} \oplus \mathbf{v}\|_0$. With originals $\{\mathbf{m}_i^{\text{orig}}\}_{i=1}^{N_o} \subset \{0,1\}^B$ and distortions $\{\mathbf{m}_j^{\text{dist}}\}_{j=1}^{N_d}$, we report:

$$\text{Intra-Orig} = \frac{2}{N_o(N_o-1)} \sum_{1 \le i < j \le N_o} \text{Ham}(\mathbf{m}_i^{\text{orig}}, \mathbf{m}_j^{\text{orig}}),$$

$$\text{Ref-vs-Dist} = \frac{1}{N_d} \sum_{j=1}^{N_d} \text{Ham}(\mathbf{m}_1^{\text{orig}}, \mathbf{m}_j^{\text{dist}}).$$

(4)

Let $\mathcal{M} = \{\mathbf{m}_i^{\text{orig}}\}_{i=1}^{N_o} \cup \{\mathbf{m}_j^{\text{dist}}\}_{j=1}^{N_d}$ and enumerate $\mathcal{M} = \{\mathbf{m}_a\}_{a=1}^N$ with $N=N_o+N_d=20$. Then

$$\text{All-20} = \frac{2}{N(N-1)} \sum_{1 \le a < b \le N} \text{Ham}(\mathbf{m}_a, \mathbf{m}_b).$$

(5)

Bit entropy uses per-bit frequency $p_k = \frac{1}{N} \sum_{a=1}^{N} m_{a,k}$:

$$\text{Entropy} = \frac{1}{B} \sum_{k=1}^{B} \left[ -p_k \log_2 p_k - (1 - p_k) \log_2(1 - p_k) \right].\tag{6}$$

For cross-prompt statistics, draw codes $\{\mathbf{c}_t\}_{t=1}^{M}$ from different prompts and set $D_{it} = \text{Ham}(\mathbf{m}_i^{\text{orig}}, \mathbf{c}_t)$; we report $\min_{i,t} D_{it}$, $\frac{1}{N_o M} \sum_{i,t} D_{it}$, and $\max_{i,t} D_{it}$. Table 12 shows strong within-prompt invariance and near-random across-prompt behavior on COCO (mean cross-prompt $\approx 509$), while SDP is weaker (mean $\approx 409$), matching our COCO-skewed training mix.

*Table 12.* Semantic masker invariance/separability on COCO vs. SDP (SD-2.1, Guidance Scale 7.5, $B=1024$; averages over 1,000 prompts).

| Dataset | Intra-Orig↓ | Ref-vs-Dist↓ | All-20↓ | Bit Ent. | Max dist↑ | Min dist↑ | Mean dist↑ | Entropy (diff-prompt)↑ |
|---------|------------|-------------|---------|----------|-----------|-----------|------------|------------------------|
| COCO | 73.47 | 22.99 | 61.50 | 0.131 | 573.02 | 319.07 | 508.77 | 0.995 |
| SDP | 102.77 | 35.92 | 86.59 | 0.203 | 552.84 | 237.82 | 428.68 | 0.810 |

**Ablation on two-stage epochs.** We vary the number of epochs in Stage 1 (`epoch1`) and Stage 2 (`epoch2`) while keeping all other settings fixed. Metrics are averaged over 500 prompts; lower is better for *Intra-Orig*, *Ref-vs-Dist*, and *All-20*. The result is shown in Table 13.

The best setting is **epoch1=180**, **epoch2=160**, which achieves the lowest values across all three distance metrics and the lowest bit entropy. Increasing Stage 2 beyond $\sim 160$ epochs tends to degrade both same-prompt consistency and distortion robustness (e.g., 180/180, 180/200).

*Table 13.* Epoch ablation. Results averaged over 500 prompts. Lower is better for the first three columns.

| Ep1 | Ep2 | Intra-Orig↓ | Ref-vs-Dist↓ | All-20↓ | Bit Ent. |
|-----|-----|-------------|--------------|---------|----------|
| 160 | 160 | 71.98 | 23.23 | 60.43 | 0.129 |
| 160 | 180 | 71.03 | 24.79 | 61.05 | 0.130 |
| 160 | 200 | 71.66 | 22.12 | 60.00 | 0.128 |
| 170 | 160 | 73.54 | 22.60 | 61.53 | 0.131 |
| 170 | 190 | 72.22 | 22.70 | 59.67 | 0.128 |
| **180** | **160** | **66.92** | **21.83** | **56.61** | **0.121** |
| 180 | 180 | 76.64 | 24.34 | 63.92 | 0.136 |
| 180 | 200 | 69.72 | 25.20 | 60.92 | 0.130 |
| 200 | 250 | 69.13 | 22.95 | 59.07 | 0.126 |

# D. Watermark Removal Attacks

We further evaluate whether the semantic masker in SemBind would make the resulting watermark more vulnerable to advanced watermark removal attacks. In particular, we consider CtrlRegen (Liu et al., 2025), a recent regeneration-based removal attack that aims to remove watermarks while preserving the image semantics.

We conduct experiments on COCO using Stable Diffusion 2.1 Base, following the same settings as in the main paper. We evaluate two representative latent-based watermarking schemes, Tree-Ring (TR) and Gaussian Shading (GS), together with their SemBind-enhanced variants, TR-S and GS-S. The evaluation is conducted on 100 prompts. We report the detection rate (Det.) and, when applicable, the bit accuracy (Acc.) before and after applying CtrlRegen.

As shown in Table 14, CtrlRegen has very similar removal effects on the SemBind variants and their corresponding base watermarking schemes. For TR, CtrlRegen substantially reduces the detection rate for both TR and TR-S, with nearly identical performance after the attack. For GS, both GS and GS-S remain largely robust after CtrlRegen, and the detection rate and bit accuracy of GS-S are close to those of GS.

These results indicate that the semantic masker does not noticeably amplify the effectiveness of CtrlRegen. This is because CtrlRegen is designed to preserve the image semantics while removing the watermark. As a result, the semantic mask

*Table 14.* Robustness against the CtrlRegen watermark removal attack on COCO with Stable Diffusion 2.1 Base. "Det." denotes the detection rate, and "Acc." denotes the bit accuracy.

| Method | Before CtrlRegen Det. / Acc. | After CtrlRegen Det. / Acc. |
|---|---|---|
| TR | 1.00 / – | 0.23 / – |
| TR-S | 1.00 / – | 0.22 / – |
| GS | 1.00 / 1.000 | 0.98 / 0.882 |
| GS-S | 0.99 / 0.998 | 0.96 / 0.879 |

recomputed from the attacked image remains close to that of the original image, and the robustness under removal attacks is mainly determined by the underlying watermarking scheme rather than by the semantic binding mechanism itself.

# E. Adaptive Attack

## E.1. Training a surrogate semantic masker.

SemBind assumes the semantic masker is kept private by the service provider. Nevertheless, an adaptive adversary may attempt to approximate this component by collecting multiple watermarked images generated under the same prompt and training a surrogate masker.

**Threat model.** We consider a deliberately *stronger-than-realistic* adaptive attacker. The adversary is assumed to possess completely **the same** training dataset(Appendix A), network architecture, loss functions, and training hyperparameters (Sec. 3.1 and Appendix B) as the defender. The only difference is the randomness in training (e.g., random seed used in training). The adversary's goal is to train a surrogate semantic masker whose outputs match the provider's codes for images generated from the same prompt.

**Experimental setup.** Following the same training recipe as in the main paper, we independently train five semantic maskers (M1–M5) using different random seeds. Each masker outputs a $B=1024$-bit semantic code. We evaluate code consistency on both COCO and SDP by randomly sampling 1000 prompts, generating one image per prompt, and extracting semantic codes using each trained masker. For each pair of maskers, we compute the Hamming distance between their codes over the same set of generated images.

**Results and discussion.** Tables 15 and 16 report the resulting pairwise distance matrices. We can see that, even under this stringent setting, independently trained maskers exhibit near-random bitwise mismatch: pairwise mean Hamming distances concentrate around $\approx 512$ bits (i.e., $\approx 50\%$ of 1024 bits) on both COCO and SDP. This suggests that training a surrogate masker does not reliably reproduce the provider's bit-level code mapping, making it difficult to directly clone the semantics-to-mask binding used by SemBind. A key reason is that our objective is primarily relational—it enforces that same-prompt samples are close and different-prompt samples are separated—and is non-identifiable up to symmetry transformations (e.g., bit permutations/flips or rotations of intermediate features), under which the loss remains (nearly) unchanged. Consequently, different random seeds can converge to different but equally valid solutions that preserve relative structure while inducing different bitwise representations.

*Table 15.* COCO: mean Hamming distance (1024-bit) between independently trained maskers (1000 prompts).

|  | M1 | M2 | M3 | M4 | M5 |
|---|---|---|---|---|---|
| M1 | 0.0 | 508.0 | 492.8 | 505.2 | 511.6 |
| M2 | 508.0 | 0.0 | 514.3 | 506.7 | 520.1 |
| M3 | 492.8 | 514.3 | 0.0 | 509.5 | 503.8 |
| M4 | 505.2 | 506.7 | 509.5 | 0.0 | 515.4 |
| M5 | 511.6 | 520.1 | 503.8 | 515.4 | 0.0 |

*Table 16.* SDP: mean Hamming distance (1024-bit) between independently trained maskers (1000 prompts).

|  | M1 | M2 | M3 | M4 | M5 |
|---|---|---|---|---|---|
| M1 | 0.0 | 510.2 | 498.7 | 507.9 | 513.4 |
| M2 | 510.2 | 0.0 | 516.8 | 509.1 | 518.6 |
| M3 | 498.7 | 516.8 | 0.0 | 511.0 | 505.6 |
| M4 | 507.9 | 509.1 | 511.0 | 0.0 | 516.2 |
| M5 | 513.4 | 518.6 | 505.6 | 516.2 | 0.0 |

**End-to-end forgery with surrogate maskers.** The above results show that independently trained maskers produce substantially different bit-level codes. We further verify whether such surrogate maskers can nevertheless be used in an end-to-end forgery attack. Specifically, we consider GS-S as the representative SemBind-enabled watermarking scheme,

where Gaussian Shading (GS) is used as the underlying latent watermark. The verifier uses the provider's deployed semantic masker to unbind and extract the watermark, while the attacker replaces this masker with an independently trained surrogate masker when performing the binding operation.

We evaluate this attack on 100 COCO prompts under the same Stable Diffusion 2.1 Base setting as in the main experiments. For each surrogate masker, the attacker embeds its own GS watermark using the surrogate semantic mask and then submits the resulting image to the GS-S verifier. We report the detection rate (Det.) and bit accuracy (Acc.) in Table 17.

*Table 17.* End-to-end forgery attempts on GS-S using independently trained surrogate semantic maskers. The verifier uses the provider's deployed masker, while the attacker performs binding with a surrogate masker.

| Surrogate Masker | Det. | Acc. |
| --- | --- | --- |
| M1 | 0.00 | 0.5104 |
| M2 | 0.00 | 0.5072 |
| M3 | 0.00 | 0.4988 |
| M4 | 0.00 | 0.5046 |
| M5 | 0.00 | 0.5089 |

As shown in Table 17, none of the surrogate maskers can produce forged images that pass the GS-S verifier. The detection rate remains $0.00$ for all five surrogate maskers, and the bit accuracy stays close to $0.5$, which is essentially random guessing. This end-to-end result confirms that the surrogate masker does not provide an effective substitute for the provider's deployed masker. Intuitively, because the attacker's surrogate mask differs from the verifier's true mask in roughly half of the bit positions, the unbinding operation at verification time randomizes the recovered GS payload, causing the forgery attempt to fail.

Overall, these results suggest that even an adaptive attacker with the same architecture, training data, loss functions, and hyperparameters cannot reliably clone the provider's semantic binding function merely by retraining a surrogate masker with a different random seed. This supports our threat model against black-box and surrogate-based adaptive attackers. We note, however, that full white-box access to the exact deployed masker remains outside our threat model.

### E.2. Pixel-level spoofing of the semantic masker

Beyond training a surrogate masker, an adaptive adversary may attempt to *spoof* the semantic masker directly in the pixel domain. For example, given a benign *source* image (e.g., a cat photo), the adversary may take an arbitrary *target* image (e.g., containing policy-violating or otherwise illicit content) and overlay the source image onto the target at some scale. This raises a natural question: does the resulting semantic code become entirely different, partially overlap with the source code, or approach the source code as the overlay grows? Equivalently, how much pixel-level overlap is required to cause the semantic masking process to be misled?

**Threat model.** We consider a spoofing attacker who modifies images in the pixel space. In this setting, the adversary is given a benign reference image and attempts to overlay it onto a chosen target image with a controllable size ratio, with the goal that the composite image yields a semantic code close to that of the benign image.

**Experimental setup.** We evaluate this spoofing attack on *both* COCO and SDP prompt sets, sampling 100 prompts from each dataset (one generated image per prompt). For the benign source image, we use a fixed "cat" image generated by Stable Diffusion. For target images, we randomly sample prompts from the COCO and SDP prompt-evaluation sets and generate one image per prompt, then extract their semantic codes using the semantic masker. We then perform pixel-space overlay by resizing the cat image to a scale ratio $r \in \{0.0, 0.1, \ldots, 1.0\}$ relative to the target image resolution and pasting it onto the target under two placements: (i) a bottom-right overlay, and (ii) a centered overlay. For each $r$, we compute the semantic code of the resulting composite image and measure its Hamming distance to the reference code at $r = 1.0$ (i.e., when the image is fully replaced by the cat image), reporting distances for $r = 0.0, \ldots, 0.9$.

**Results and discussion.** Results are summarized in Figs. 6a and 6b, with an illustrative example shown in Fig. 5. Overall, increasing the cat overlay ratio $r$ monotonically reduces the Hamming distance to the source code (at $r = 1.0$), confirming that large pixel-level overlap can pull the semantic code toward the benign source. However, the effect is limited for moderate overlay sizes: across both COCO and SDP, the semantic code remains far from the source for $r \leq 0.8$, and a sharp transition only occurs when the overlay becomes extremely large (around $r = 0.9$, close to fully replacing the target image).

This indicates that SemBind largely preserves its anti-forgery effectiveness against pixel-space spoofing unless the adversary is willing to overwrite most of the target content with benign pixels. In practical forgery scenarios, such a large benign overlay would substantially compromise the attacker's intended (illicit) semantics and thus significantly reduce the utility of the forged image. We therefore conclude that pixel-level spoofing is an ineffective strategy to neutralize SemBind in practice, as successfully misleading the semantic masker requires overwhelming the target image with benign content.

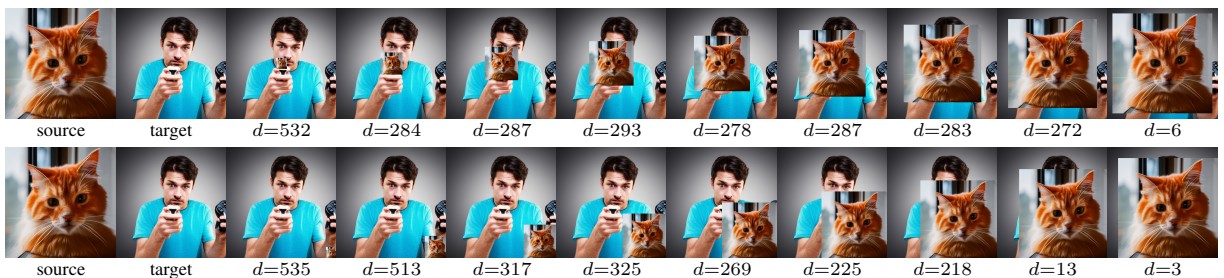

Figure 5. Pixel-level spoofing by overlaying a benign source image (cat) onto a target image at scale ratio $r \in \{0.1, 0.2, \ldots, 0.9\}$ (with $r = 0.0$ the original target and $r = 1.0$ the full source image). Top: centered overlay. Bottom: bottom-right overlay. Labels report the Hamming distance $d$ to the reference semantic code of the source image.

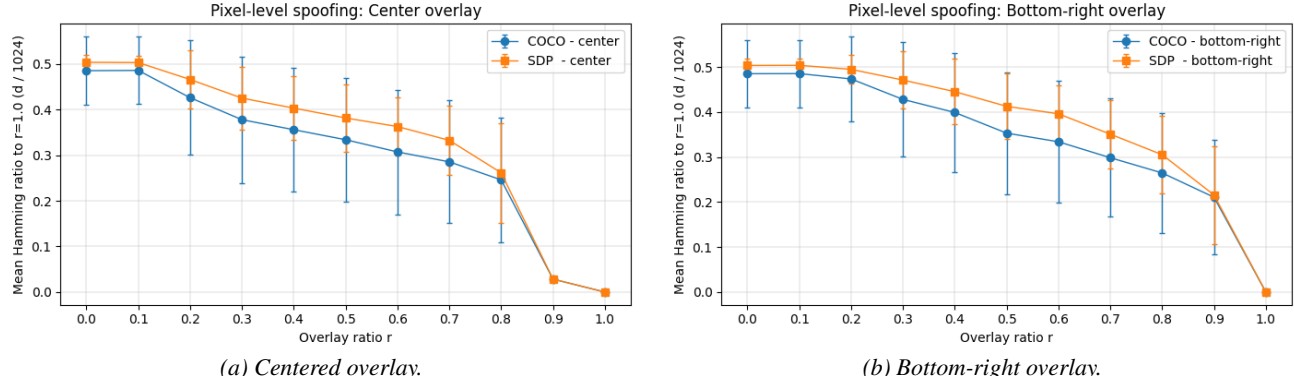

*(a) Centered overlay.*                           *(b) Bottom-right overlay.*

Figure 6. Pixel-level spoofing curves on COCO and SDP (100 prompts each). We report the mean Hamming distance (normalized by 1024) between the semantic code of the composite image at overlay ratio $r$ and the reference code at $r = 1.0$ (full cat image). Error bars indicate $\pm 1$ standard deviation.

## F. Generalization Experiments

To further demonstrate the generalization of SemBind, we conduct two additional experiments. First, we evaluate SemBind on the same image generation model under a more realistic and broader prompt dataset in F.1, assessing robustness and defense against forgery attacks beyond our primary evaluation sets. Second, we study the feasibility of applying SemBind to FLUX (Labs et al., 2025) in F.2, testing whether the proposed semantic binding mechanism transfers to a different diffusion-based generator. We next describe the experimental setups and then report the results with analysis for each experiment.

### F.1. Generalization on a Larger Prompt Dataset

**Experimental setup.** To evaluate generalization, we reuse the same SemBind-enabled variants and hyperparameter settings as in the main paper (Sec. 4.1), and repeat the forgery-resistance and robustness evaluations on a larger and more realistic prompt corpus, `FredZhang7/stable-diffusion-prompts-2.47M`[8]. Unless stated otherwise, all watermark backends, detection thresholds (FPR $= 10^{-6}$), and mask ratios are identical to the main paper.

---

[8]https://huggingface.co/datasets/FredZhang7/stable-diffusion-prompts-2.47M

**Black-box forgery attacks.** We follow the same black-box forgery experiment setting as Sec. 4.3, and consider the two canonical attacks of Müller *et al.* (Müller et al., 2025): *imprinting* and *reprompting*. As attacker models, we again use Stable Diffusion v2.1 (match) and Stable Diffusion v1.5 (mismatch).

**Imprinting.** For each watermarking scheme and attacker model, we sample 100 prompts from the 2.47M prompt corpus and generate 100 corresponding watermarked images. As target cover images, we randomly sample 100 natural photographs from the MS-COCO validation set. We then run the imprinting optimization exactly as in (Müller et al., 2025) and Sec. 4.3 (150 gradient steps with learning rate 0.01), and probe watermark detection every 50 steps.

Results are reported in Table 18. Overall, SemBind continues to provide strong protection against imprinting on this larger and more realistic prompt corpus. In particular, imprinting is largely ineffective against TR-S and GS-S, with detection rates remaining near zero throughout optimization in both the match (SD 2.1) and mismatch (SD 1.5) settings. For PRC and GS++, SemBind also substantially reduces the attack success rate, yielding markedly lower detection/bit-recovery performance compared to the corresponding baselines across all probed steps.

*Table 18.* Imprint forgery attack on the FredZhang7/stable-diffusion-prompts-2.47M dataset for four latent-based watermarking schemes and their SemBind-enhanced variants.

| Method | Step | Attacker Model: SD 2.1 | | | Attacker Model: SD 1.5 | | |
| | | Det.↓ | Bit Acc.↓ | PSNR | Det.↓ | Bit Acc.↓ | PSNR |
| --- | --- | --- | --- | --- | --- | --- | --- |
| TR | 50/100/150 | 1.00/1.00/1.00 | — | 23.37/22.21/21.25 | 1.00/1.00/1.00 | — | 22.86/21.83/21.15 |
| TR-S | 50/100/150 | 0.02/0.03/0.03 | — | 23.32/22.12/21.35 | 0.01/0.02/0.02 | — | 22.87/21.83/21.15 |
| GS | 50/100/150 | 1.00/1.00/1.00 | 1.0000/1.0000/1.0000 | 23.35/22.15/21.39 | 1.00/1.00/1.00 | 0.9990/0.9996/0.9998 | 22.88/21.85/21.18 |
| GS-S | 50/100/150 | 0.05/0.05/0.07 | 0.4901/0.4926/0.5007 | 23.33/22.13/21.36 | 0.02/0.03/0.04 | 0.4936/0.4951/0.4902 | 22.89/21.84/21.16 |
| PRC | 50/100/150 | 1.00/1.00/1.00 | 1.0000/1.0000/1.0000 | 23.36/22.16/21.39 | 1.00/1.00/1.00 | 1.0000/1.0000/1.0000 | 22.91/21.88/21.21 |
| PRC-S | 50/100/150 | 0.08/0.20/0.24 | 0.5402/0.5631/0.5210 | 23.38/22.18/21.41 | 0.01/0.02/0.02 | 0.5068/0.5112/0.5115 | 22.92/21.89/21.20 |
| GS++ | 50/100/150 | 1.00/1.00/1.00 | 0.9995/0.9999/1.0000 | 23.36/22.17/21.41 | 0.99/0.99/1.00 | 0.9803/0.9881/0.9944 | 22.90/21.87/21.18 |
| GS++-S | 50/100/150 | 0.18/0.38/0.51 | 0.5820/0.6845/0.7423 | 23.35/22.20/21.38 | 0.03/0.05/0.09 | 0.5125/0.5298/0.5469 | 22.90/21.87/21.19 |

**Reprompting.** For each scheme, we sample 100 prompts from the 2.47M corpus to generate watermarked images, and sample an additional 100 mismatched prompts from the I2P dataset[9] for reprompting. The adversary reuses the estimated watermarked initial latent and runs the proxy model forward under the mismatched prompts, following the same procedure as in Sec. 4.3.

Results are summarized in Table 19. Overall, SemBind remains highly effective against reprompting on this larger prompt corpus: for all four backends, the SemBind-enabled variants exhibit substantially reduced forgery success compared to their corresponding baselines, under both the match (SD 2.1) and mismatch (SD 1.5) attacker models.

*Table 19.* Reprompt forgery attack for four latent-based watermarking schemes and their SemBind-enhanced variants (evaluated on FredZhang7/stable-diffusion-prompts-2.47M dataset).

| Method | SD 2.1 attacker | | SD 1.5 attacker | |
| | Det.↓ | Bit Acc.↓ | Det.↓ | Bit Acc.↓ |
| --- | --- | --- | --- | --- |
| TR | 1.00 | — | 0.99 | — |
| TR-S | 0.46 | — | 0.44 | — |
| GS | 0.99 | 0.9876 | 0.99 | 0.9884 |
| GS-S | 0.56 | 0.6228 | 0.50 | 0.6059 |
| PRC | 0.95 | 0.9723 | 0.93 | 0.9692 |
| PRC-S | 0.53 | 0.7421 | 0.13 | 0.5715 |
| GS++ | 0.90 | 0.9461 | 0.87 | 0.8824 |
| GS++-S | 0.45 | 0.7132 | 0.16 | 0.5756 |

**Robustness.** We follow the same robustness experiment settings as in Sec. 4.4.

The results in Table 20 show that SemBind generalizes well to this larger prompt corpus: although most prompts in `FredZhang7/stable-diffusion-prompts-2.47M` do not appear in the data used to train the semantic masker,

---

[9]https://huggingface.co/datasets/AIML-TUDA/i2p

the SemBind-enabled variants retain strong robustness under most distortions. As in the main paper, PRC and GS++ exhibit larger robustness drops under the stronger filtering distortions (GauBlur and MedFilter). We attribute this primarily to the limited strength/frequency of such filtering-style augmentations during semantic-masker training; in principle, this can be mitigated by adjusting the augmentation recipe or further fine-tuning the masker on augmentations that better match these distortions.

*Table 20.* Robustness under common distortions on `FredZhang7/stable-diffusion-prompts-2.47M`. "Average (Distortion)" is the mean across the six distortion types (excluding "None").

| Method | None (Det./Acc.) | JPEG (QF=70) | Brightness | GauBlur ($r$=3) | GauNoise ($\sigma$=0.01) | MedFilter ($k$=7) | Resize (×0.5) | Average (Distortion) |
|---|---|---|---|---|---|---|---|---|
| TR | 1.00 | 0.97 | 1.00 | 1.00 | 1.00 | 1.00 | 1.00 | 0.995 |
| TR-S | 1.00 | 0.90 | 0.97 | 0.96 | 0.93 | 0.93 | 0.96 | 0.942 |
| GS | 1.00/1.0000 | 1.00/1.0000 | 1.00/0.9987 | 1.00/0.9961 | 1.00/0.9939 | 1.00/0.9982 | 1.00/0.9997 | 1.000/0.9978 |
| GS-S | 1.00/0.9873 | 0.98/0.9741 | 0.98/0.9804 | 0.99/0.9547 | 0.99/0.9593 | 0.99/0.9871 | 1.00/0.9803 | 0.988/0.9726 |
| PRC | 1.00/1.0000 | 1.00/1.0000 | 1.00/1.0000 | 0.94/0.9746 | 0.99/0.9950 | 0.94/0.9748 | 1.00/1.0000 | 0.978/0.9907 |
| PRC-S | 1.00/1.0000 | 0.98/0.9854 | 0.95/0.9598 | 0.71/0.8802 | 0.95/0.9748 | 0.75/0.8782 | 0.98/0.9950 | 0.887/0.9456 |
| GS++ | 1.00/0.9998 | 0.98/0.9913 | 0.96/0.9841 | 0.97/0.9866 | 0.85/0.9259 | 0.94/0.9725 | 1.00/1.0000 | 0.950/0.9767 |
| GS++-S | 1.00/0.9998 | 0.96/0.9822 | 0.94/0.9691 | 0.86/0.9344 | 0.82/0.9075 | 0.80/0.9019 | 0.97/0.9856 | 0.892/0.9468 |

## F.2. Generalization on FLUX

**Experimental setup.**   To demonstrate that SemBind can transfer to other image generation models, we further evaluate it on FLUX. To balance feasibility and computational cost, we focus on Gaussian Shading (GS) and its SemBind-enabled variant (GS-S), and deploy them on FLUX 1.dev. We reuse the same trained semantic masker (without retraining) and integrate it with GS in the FLUX pipeline. All experiments are conducted at $512\times512$ resolution, where the latent has shape $1\times16\times64\times64$, and we embed a 256-bit payload. We then evaluate forgery resistance and robustness on both the SDP and COCO prompt sets.

**Black-box forgery attacks.**   We follow the same black-box forgery experiment settings as the main paper, and consider the two canonical attacks of Müller *et al.* (Müller et al., 2025): *imprinting* and *reprompting*.

We use Stable Diffusion v2.1 as the attacker model; note that this corresponds to the mismatch case for FLUX.

**Robustness.**   We follow the same robustness protocol as in the Sec. 4.4. We evaluate GS and its SemBind-enabled variant (GS-S) on FLUX 1.dev using 100 prompts from each of the COCO and SDP prompt sets.

Results are reported in Table 21. Overall, SemBind transfers to FLUX without introducing notable robustness issues: GS-S maintains high detection/bit-accuracy under all tested distortions, with only a modest drop relative to the GS baseline.

*Table 21.* FLUX robustness under common distortions. Rows 1–2 report COCO results and rows 3–4 report SDP results. "Average (Distortion)" is the mean across the six distortion types (excluding "None").

| Method | None (Det./Acc.) | JPEG (QF=70) | Brightness | GauBlur ($r$=3) | GauNoise ($\sigma$=0.01) | MedFilter ($k$=7) | Resize (×0.5) | Average (Distortion) |
|---|---|---|---|---|---|---|---|---|
| GS | 1.0/1.0000 | 0.99/0.9789 | 0.99/0.9928 | 1.0/0.9911 | 0.99/0.9937 | 1.0/0.9937 | 1.0/0.9988 | 0.995/0.9915 |
| GS-S | 1.0/0.9989 | 1.0/0.9625 | 1.0/0.9939 | 1.0/0.9668 | 1.0/0.9884 | 1.0/0.9816 | 1.0/0.9938 | 1.000/0.9812 |
| GS | 1.0/0.9995 | 0.99/0.9933 | 1.0/0.9966 | 1.0/0.9894 | 0.99/0.9954 | 1.0/0.9909 | 1.0/0.9977 | 0.997/0.9939 |
| GS-S | 0.98/0.9872 | 0.98/0.9592 | 0.98/0.9867 | 0.98/0.9272 | 0.97/0.9757 | 0.98/0.9300 | 1.0/0.9864 | 0.982/0.9609 |

**Imprinting.**   We sample 100 prompts from each of the SDP and COCO prompt sets and generate 100 corresponding watermarked images with FLUX 1.dev. As target cover images, we randomly sample 100 natural photographs from the MS-COCO validation set. We then run the imprinting optimization for 150 gradient steps with learning rate 0.01, and probe watermark detection every 50 steps.

Results are reported in Table 22. Notably, although the attacker uses Stable Diffusion v2.1 as the proxy model (a mismatch setting with a substantial gap to FLUX), the baseline GS still achieves over 80% forgery success. In contrast, the SemBind-enabled variant (GS-S) fully suppresses imprinting across all probed steps, demonstrating that SemBind transfers effectively to a different generator backbone and provides strong anti-forgery generalization.

*Table 22.* FLUX imprinting attack results.

| Method | COCO | | | | SDP | | | |
|--------|------|------|----------|------|------|------|----------|------|
| | Step | Det.↓ | Bit Acc.↓ | PSNR | Step | Det.↓ | Bit Acc.↓ | PSNR |
| GS | 50/100/150 | 0.86/0.91/0.94 | 0.7566/0.8930/0.9254 | 23.4399 | 50/100/150 | 0.80/0.90/0.92 | 0.7629/0.8730/0.9174 | 23.4796 |
| GS-S | 50/100/150 | 0.00/0.00/0.00 | 0.5020/0.5000/0.5039 | 23.4579 | 50/100/150 | 0.00/0.00/0.00 | 0.4875/0.4832/0.4934 | 23.4622 |

**Reprompting.**    We sample 100 prompts from each of the SDP and COCO prompt sets to generate watermarked images, and sample an additional 100 mismatched prompts from the I2P dataset for the reprompting attack. The adversary reuses the estimated watermarked initial latent and runs the SD 2.1 proxy model forward under the mismatched prompts, following the same procedure as in the main paper.

Results are reported in Table 23. Overall, SemBind remains effective against reprompting on FLUX: compared to the baseline GS, the SemBind-enabled variant (GS-S) substantially reduces forgery success across both COCO and SDP.

*Table 23.* FLUX reprompting attack results.

| Method | COCO | | SDP | |
|--------|------|----------|------|----------|
| | Det.↓ | Bit Acc.↓ | Det.↓ | Bit Acc.↓ |
| GS | 0.47 | 0.6453 | 0.78 | 0.7191 |
| GS-S | 0.01 | 0.5039 | 0.00 | 0.5070 |

# G. Additional Black-Box Forgery Attacks

We further evaluate SemBind against WMCopier (Dong et al., 2025), a recent black-box watermark forgery attack that is structurally different from the imprinting and reprompting attacks considered in the main experiments. WMCopier trains an attacker-side diffusion model to imitate the watermark distribution from a set of watermarked images, and then uses the trained model to forge watermarks on arbitrary images.

**Experimental setup.**    We conduct experiments on COCO using Stable Diffusion 2.1 Base. Following the setting of WMCopier, we fix the watermark key for each method. For each watermarking scheme, we generate 1,000 watermarked images with different prompts to train the attacker's diffusion model, and train it for 10,000 steps. At test time, we generate 100 clean images for forgery and evaluate whether the forged images can pass the corresponding watermark detector. We test Tree-Ring (TR) and Gaussian Shading (GS), together with their SemBind variants, TR-S and GS-S. We report the detection rate (Det.) and, when applicable, the bit accuracy (Acc.).

*Table 24.* Black-box forgery results under WMCopier on COCO with Stable Diffusion 2.1 Base. Lower Det. / Acc. indicates stronger resistance to forgery.

| Method | Det. | Acc. |
|--------|------|------|
| TR | 0.66 | – |
| TR-S | 0.02 | – |
| GS | 0.13 | 0.6742 |
| GS-S | 0.00 | 0.5495 |

**Results and discussion.**    As shown in Table 24, WMCopier can forge the original TR watermark with a detection rate of 0.66, and also increases the extracted accuracy of GS above random guessing. In contrast, the SemBind variants remain highly resistant to this attack: TR-S reduces the detection rate to 0.02, while GS-S achieves a detection rate of 0.00 and a bit accuracy close to random guessing.

These results show that SemBind substantially improves resistance to WMCopier. A likely reason is that, under a fixed key, the original TR and GS schemes induce relatively fixed latent watermark distributions that can be imitated by the attacker-side model. By contrast, SemBind binds the watermark to prompt-dependent image semantics, producing more diverse latent distributions and making the watermark much harder to copy across unrelated images.

# H. Proof of Theorem 4.1

**Notation.** Let $L = CHW$ be the latent dimension. Write $\mathcal{N} \equiv \mathcal{N}(0, I_L)$ for the $L$-dimensional standard Gaussian, and $\mathcal{N}^{\otimes t}$ for $t$ i.i.d. copies. For a sign mask $S \in \{\pm 1\}^L$ and a permutation $\pi$ of $[L]$, denote by

$$F_{S,\pi}(u) = P_\pi \operatorname{Diag}(S)\, u \quad \text{for } u \in \mathbb{R}^L,$$

where $P_\pi$ is the permutation matrix. We allow $(S, \pi)$ to be jointly sampled (and even shared across $t$ samples), but require $(S, \pi)$ be independent of the challenge latents.

**Definition H.1** (Single-/multi-sample undetectability). A watermarking scheme $\mathcal{W}$ (on initial latents) is **single-sample** $(\varepsilon, \mathsf{Adv})$-**undetectable** if for every (non-uniform) adversary $D \in \mathsf{Adv}$, satisfying:

$$\big| \Pr[D(Z) = 1] - \Pr[D(G) = 1] \big| \leq \varepsilon, \tag{7}$$

with $Z \leftarrow \mathcal{W}$, $G \leftarrow \mathcal{N}$.

It is **multi-sample** $(t, \varepsilon, \mathsf{Adv})$-**undetectable** if for every $D \in \mathsf{Adv}$, satisfying:

$$\big| \Pr[D(Z_1, \ldots, Z_t) = 1] - \Pr[D(G_1, \ldots, G_t) = 1] \big| \leq \varepsilon, \tag{8}$$

with $(Z_i)_{i=1}^t \xleftarrow{\text{i.i.d.}} \mathcal{W}$ and $(G_i)_{i=1}^t \xleftarrow{\text{i.i.d.}} \mathcal{N}$. Here $\mathsf{Adv}$ may be any probabilistic polynomial-time (PPT) adversary (Goldwasser & Micali, 1984; Goldreich, 2001).

**Definition H.2** (SemBind post-processing on initial latents). Given a base scheme $\mathcal{W}$ that outputs $Z \leftarrow \mathcal{W}$, its SemBind variant first samples randomness $R$ (independent of $Z$) which determines a sign mask $S \in \{\pm 1\}^L$ and a permutation $\pi$ (both possibly correlated across samples and possibly encoding a mask ratio by forcing some entries of $S$ to $+1$), and outputs

$$Z^{\text{sem}} = F_{S,\pi}(Z).$$

**Lemma H.3** (Gaussian invariance under independent sign flips and permutations). *Let $G \leftarrow \mathcal{N}$ and $(S, \pi) \perp G$. Then $F_{S,\pi}(G) \overset{d}{=} G$. The same holds for $t$ i.i.d. copies jointly transformed by $\{(S_i, \pi_i)\}_{i=1}^t$ independent of $\{G_i\}_{i=1}^t$.*

*Proof.* For fixed $(S, \pi)$, the map $u \mapsto P_\pi \operatorname{Diag}(S) u$ permutes coordinates and flips signs, hence preserves $\mathcal{N}(0, I_L)$. Independence allows averaging over $(S, \pi)$. $\square$

**Lemma H.4** (Closure under independent post-processing). *Let $X, Y$ be random variables on $\mathbb{R}^L$, and $T$ any (possibly randomized) map independent of $(X, Y)$. Then for any distinguisher class $\mathsf{Adv}$ and $\varepsilon \geq 0$,*

$$\Delta_{\mathsf{Adv}}\big(T(X), T(Y)\big) \leq \Delta_{\mathsf{Adv}}(X, Y), \tag{9}$$

*where $\Delta_{\mathsf{Adv}}(U, V) = \sup_{D \in \mathsf{Adv}} |\Pr[D(U) = 1] - \Pr[D(V) = 1]|$.*

*Proof.* For fixed $D$, define $D_T(u) = \mathbb{E}[D(T(u))]$ over $T$'s randomness. Then $|\Pr[D(T(X)) = 1] - \Pr[D(T(Y)) = 1]| = |\Pr[D_T(X) = 1] - \Pr[D_T(Y) = 1]| \leq \Delta_{\mathsf{Adv}}(X, Y)$. $\square$

**Theorem H.5** (Semantic masking preserves undetectability). *If $\mathcal{W}$ is single-sample $(\varepsilon, \mathsf{Adv})$-undetectable (resp. multi-sample $(t, \varepsilon, \mathsf{Adv})$-undetectable), then its SemBind variant $\mathcal{W}^{\text{sem}}$ per Def. H.2 is single-sample (resp. multi-sample) $(\varepsilon, \mathsf{Adv})$-undetectable as well.*

*Proof.* **Single-sample.** Let $Z \leftarrow \mathcal{W}$, $G \leftarrow \mathcal{N}$, and $T(\cdot) = F_{S,\pi}(\cdot)$ with $(S, \pi) \perp (Z, G)$. By Lemma H.4,

$$\Delta_{\mathsf{Adv}}\big(F_{S,\pi}(Z),\, F_{S,\pi}(G)\big) \leq \Delta_{\mathsf{Adv}}(Z, G) \leq \varepsilon. \tag{10}$$

By Lemma H.3, $F_{S,\pi}(G) \overset{d}{=} G$, hence $\Delta_{\mathsf{Adv}}\big(F_{S,\pi}(Z), G\big) \leq \varepsilon$.

**Multi-sample.** Let $(Z_i)_{i=1}^t \xleftarrow{\text{i.i.d.}} \mathcal{W}$, $(G_i)_{i=1}^t \xleftarrow{\text{i.i.d.}} \mathcal{N}$, and $T((u_i)_{i=1}^t) = (F_{S_i,\pi_i}(u_i))_{i=1}^t$ with $\{(S_i, \pi_i)\}_{i=1}^t \perp \{(Z_i, G_i)\}_{i=1}^t$. Then we have:

$$\begin{aligned} \Delta_{\mathsf{Adv}}\big((F_{S_i,\pi_i}(Z_i))_{i=1}^t,\, (F_{S_i,\pi_i}(G_i))_{i=1}^t\big) \\ \leq \Delta_{\mathsf{Adv}}\big((Z_i)_{i=1}^t, (G_i)_{i=1}^t\big) \leq \varepsilon. \end{aligned} \tag{11}$$

and Lemma H.3 yields $(F_{S_i,\pi_i}(G_i))_{i=1}^t \overset{d}{=} (G_i)_{i=1}^t$, concluding the claim. $\square$

**Corollary H.6** (Instantiations)**.** *If Gaussian Shading is single-sample undetectable, then its SemBind variant (GS-S) remains single-sample undetectable with the same bound. If PRC or Gaussian Shading++ is multi-sample undetectable, then PRC-S and GS++-S inherit the same multi-sample bound.*

*Remark* H.7 (Scope). (1) Only independence of $(S, \pi)$ from the challenge samples is required; mask ratio (some coordinates forced to $+1$), shared/correlated masks across samples, and any secret permutation are all permitted. (2) No assumptions on bit balance or inter-bit independence are needed for undetectability (they matter for robustness/forgery resistance, not for indistinguishability).

