# OpenReview forum: "SemBind: Binding Diffusion Watermarks to Semantics Against Black-Box Forgery Attacks"
_ICML.cc/2026/Conference — ICML 2026 regular_

### Official Review · Reviewer_m3qT · 2026-03-09

**Soundness:** 3
**Presentation:** 2
**Significance:** 3
**Originality:** 3
**Overall Recommendation:** 4
**Confidence:** 3

**Summary:**

This paper proposes SemBind, a defense framework that binds latent-based diffusion watermarks to image semantics to
  counter black-box forgery attacks (imprinting and reprompting). A semantic masker trained with supervised contrastive
  learning maps images to binary codes that are near-invariant within the same prompt and near-orthogonal across
  different prompts. These codes modulate the initial watermarked latent via a sign mask, binding the watermark to
  semantic content. SemBind is compatible with any latent-based watermarking scheme, preserves image quality, and
  provides a theoretical proof of undetectability preservation. Experiments on four watermarking methods (Tree-Ring,
  Gaussian Shading, PRC, GS++) show substantial reductions in forgery success, with generalization to FLUX.

**Compliance With Llm Reviewing Policy:**

Affirmed.

**Final Justification:**

I thank the authors for their responses. My concerns are addressed during rebuttal, and I will keep my positive assessment.

**Key Questions For Authors:**

- Removal attacks are entirely unaddressed.  Watermark removal attacks are well-documented and can be effective against latent-based schemes. SemBind provides no defense
  against this threat, and the paper does not discuss it.
- Theorem 4.1 states that SemBind preserves the provable undetectability of the underlying watermarking scheme Tree-Ring, which is detectable  in principle.  How could the 'ring' disappear? These results (both theoretical and experimental) are misleading.

- SemBind is explicitly positioned not as a new watermarking scheme but as
  a "forgery defense plug-in" compatible with existing schemes. While this compatibility is a strength, it also means
  SemBind does not advance the core watermarking problem and contributes no new guarantees to the underlying security or
   robustness of these schemes.
- The paper does not discuss RingID[1] , which also attempts to distinguish watermarks
  across users by encoding identity information into the initial latent. is it possible that RingID-style approaches
   already provide partial defense against forgery as a byproduct of per-user encoding?
-  Appendix D discusses adaptive attacks. However, an adaptive adversary with access to many watermarked images from the same prompt could exploit the fact that intra-prompt Hamming distance is only ~73 bits (Table 10, COCO), potentially recovering consistent bits through
  majority voting across N watermarked samples. This multi-sample attack is not evaluated.



-  The "Brightness×1.0" distortion in Tables 5/6 applies no transformation at all and is meaningless as an
  evaluation condition. Replace with a more informative distortion like random crop.

[1] Ci H, Yang P, Song Y, et al. Ringid: Rethinking tree-ring watermarking for enhanced multi-key identification[C]//European conference on computer vision. Cham: Springer Nature Switzerland, 2024: 338-354.

**Limitations:**

yes

**Strengths And Weaknesses:**

Strengths:
  - Novel semantic binding approach that works as a plug-in layer over any latent-based watermarking scheme
  - Formal proof that provable undetectability of the underlying scheme is preserved
  - Broad compatibility validated across 4 watermarking methods and cross-model generalization to FLUX
  - No measurable image quality degradation (FID/CLIP scores statistically indistinguishable from baseline)

  Weaknesses:
 - Removal attacks are entirely unaddressed.  Watermark removal attacks are well-documented and can be effective against latent-based schemes. SemBind provides no defense against this threat, and the paper does not discuss it.
- A misalignment between SemBind's theoretical guarantee and its empirical scope. Tree Ring is detectable but you claimed your undetectable theory can be applied and the results show that Tree Ring, after SemBind become undetectable.
  - The latent-based watermarking space is already crowded; SemBind's contribution is a defense plug-in rather than a
  new watermarking paradigm, which limits its scope;
  - Adaptive attacker analysis is qualitative; no quantitative experiments against surrogate maskers
  - GS++ remains insufficiently protected under reprompting attacks (Det. up to 0.49 on SDP)
  - Auxiliary clean image generation requirement adds deployment overhead

---

> ### Author Rebuttal · Authors · 2026-03-30
>
> Thank you for the positive assessment and the constructive questions. We address the reviewer’s key concerns below.
>
> **1. Answer to Q1&W1 (Removal attack experiments).**
>
> We additionally evaluated a recent watermark removal attack, **CtrlRegen**. The results show that the removal performance of CtrlRegen on the SemBind variants is close to that of the corresponding base schemes. Thus, SemBind does not noticeably increase the success of CtrlRegen. This is because CtrlRegen aims to preserve the image semantics while removing the watermark, so the recomputed semantic mask remains similar. For details, please refer to **Rebuttal for Review P4WR, Answer to Q3**.
>
>
> **2. Answer to Q2&W2 (Clarification on Theorem 4.1 and Tree-Ring).**
>
> We believe there may be a misunderstanding regarding the scope of Theorem 4.1, so we would like to clarify it. The meaning of Theorem 4.1 is: **if the underlying watermarking scheme already has a provable undetectability guarantee, then its SemBind variant preserves the same type of undetectability guarantee.** For example, if GS is single-sample undetectable, then GS-S remains single-sample undetectable; if PRC and GS++ are multi-sample undetectable, then PRC-S and GS++-S preserve that same multi-sample undetectability.
>
> By contrast, for Tree-Ring (TR), we explicitly state in the main paper that **Tree-Ring embeds a zero-bit watermark by imposing a characteristic pattern in the initial latent and does not provide a provable undetectability guarantee** (see **Related Work, Section 2.2**). Therefore, we do not claim that TR-S is undetectable, nor do we apply Theorem 4.1 to TR in the theoretical section.
>
>
> **3. Answer to Q3 (Forgery defense plug-in).**
> We agree that SemBind is not a new watermarking paradigm, but a plug-in defense for existing latent watermarking schemes. This is intentional: our goal is not to redesign watermark embedding itself, but to address the forgery vulnerability shared by many existing methods.
>
> We view this plug-in property as a strength. SemBind can enhance multiple watermarking backbones without changing their original design, which makes it broadly applicable in practice.
>
> **4. Answer to Q4 (Discuss RingID).**
> Thank you for this suggestion. We agree that **RingID** is a relevant work and should be discussed. However, we believe RingID and SemBind address different problems. RingID mainly focuses on **multi-key identification**, i.e., distinguishing whose watermark is present, while SemBind focuses on **anti-forgery**, i.e., preventing a watermark from being transferred to semantically mismatched content.
>
> Under a black-box forgery attack, if the stolen latent watermark also contains user identity information, the forged image will typically inherit the same identity encoding as well. In other words, RingID alone does not prevent watermark forgery.
>
> **5. Answer to Q5 (Voting-based adaptive attacker).**
>
> We agree that a multi-sample attacker is a meaningful adaptive threat to consider. However, we believe a simple averaging strategy is unlikely to work against SemBind.
>
> The key reason is that, in SemBind, the semantic code and the watermarked initial latent are coupled through an element-wise Hadamard product. To successfully forge a new image, the attacker would need to remove the original semantic mask from the coupled result and then inject the semantic mask of the new target image.
>
> This therefore leads to two challenges. First, the attacker cannot remove the semantic-mask term simply by averaging samples generated from the same prompt, because these samples share the same watermark and have similar masks. Second, the attacker cannot obtain the semantic mask of the new target image because the deployed semantic masker is private to the service provider. Therefore, even if many watermarked images of the same prompt are available, the averaging-based attack does not directly enable re-binding the watermark to arbitrary new content.
>
> **6. Answer to Q6 (Inaccurate table label).**
> Thank you for pointing this out. This entry is indeed an inaccurate label in the table. It actually corresponds to ColorJitter with brightness strength = 1.0. Under this setting, the brightness factor is randomly sampled from [0, 2]. We agree that the current wording is misleading, and we will revise it in the final version.

---

> > ### Author Rebuttal · Reviewer_m3qT · 2026-04-02
> >
> > Thank you for your responses.

---

> > > ### Author Response · Authors · 2026-04-04
> > >
> > > Thank you for your response and positive assessment. We appreciate your time and consideration.

---

### Official Review · Reviewer_Hcer · 2026-03-09

**Soundness:** 3
**Presentation:** 2
**Significance:** 2
**Originality:** 2
**Overall Recommendation:** 4
**Confidence:** 4

**Summary:**

This paper proposes \textbf{SemBind}, a semantic binding enhancement for \textbf{latent-space watermarking} in diffusion models to defend against \textbf{black-box watermark forgery/spoofing}. The key idea is to train a Semantic Masker $f_\theta$ that takes an image as input and outputs a binary semantic code, which is expanded into a spatial sign mask $S\in\{-1,+1\}^{C\times H\times W}$. During embedding, SemBind modulates the (watermarked) initial latent/noise via element-wise sign flipping with $S$, thereby binding the watermark to image semantics. During extraction, the mask is recomputed from the queried image and used to ``unbind'' the latent before decoding. The intended effect is that semantic mismatch (e.g., forged images) yields significant unbinding errors and degrades verification.

**Compliance With Llm Reviewing Policy:**

Affirmed.

**Final Justification:**

My concerns have been addressed.

**Key Questions For Authors:**

1. Related Work does not sufficiently organize attacks and threat models.

2. There appears to be a large blank paragraph in the page 5.

**Limitations:**

Yes

**Strengths And Weaknesses:**

Strengths

1. SemBind is designed as an add-on wrapper over existing latent watermarking schemes, making it potentially applicable across multiple watermarking backbones without redesigning the watermark itself.

2. Black-box forgery/spoofing is a practical concern for watermark-based provenance. Binding the watermark to semantics is a reasonable direction to reduce transferability across mismatched content.

3. The ``bind/unbind with a sign mask'' construction is simple and interpretable, and it naturally yields asymmetric behavior: benign edits that preserve semantics may retain mask alignment, while semantic changes can cause large decoding disruption.

Weaknesses：

1. The evaluation primarily considers the M\"uller et al. black-box forgery attack. However, several newer spoofing/forgery strategies have appeared recently (e.g., [1], and others). Limiting evaluation to a single attack raises concerns about the generality and novelty of the defense. Evaluate against 2--3 additional recent forgery methods, including those not structurally similar to M\"uller’s pipeline, and report comparative robustness.

2. Threat model is underspecified; Semantic Masker security is not argued. The paper does not clearly define attacker capabilities (e.g., access to detector queries, access to a watermarked dataset, ability to train surrogates, knowledge of model architecture, etc.). Even with a black-box masker, an adversary may (i) \textbf{train a surrogate masker} on similar data to approximate the mask distribution; and (ii) perform \textbf{targeted optimization/adversarial examples} to steer outputs toward a desired latent mask (targeted spoofing).

3. Lack of analysis on semantic-preserving edits and prompt-distance effects. Since the mechanism relies on mask mismatch under semantic changes, a key boundary case is semantic-preserving editing (minor retouching, style transfer, local inpainting/patch replacement). In such cases, the mask may remain largely consistent, potentially allowing forged content to pass verification.

4. Missing sensitivity/ablation of key hyperparameters. SemBind introduces parameters such as mask ratio/strength (e.g., $\sigma$) and code length $B$. Larger binding strength may improve anti-forgery performance but can reduce robustness to benign transformations (JPEG, resize).

5. No quantitative link between mask mismatch and verification failure; collision/brute-force considerations missing. The paper does not proof or characterize:  (i) how small prompt paraphrases or minor image transforms perturb the sign mask,  (ii) how mask mismatch affect watermark Bit Acc/ verification Acc, and  (iii) whether tolerance thresholds enable collision-based bypass attempts.

[1] Imperceptible but forgeable: Practical invisible watermark forgery via diffusion models.

---

> ### Author Rebuttal · Authors · 2026-03-30
>
> We sincerely thank the reviewer for the constructive suggestions and address the main concerns below.
>
> **1. Answer to W1&Q1 (Additional black-box forgery attacks).**
>
> Following the reviewer’s suggestion, we additionally evaluated **WMCopier** [1] on TR, GS, TR-S, and GS-S using COCO and Stable Diffusion 2.1 Base.
>
> We fixed the watermark key. For each method, we generated 1,000 watermarked images with different prompts to train the attacker’s diffusion model, ran 10,000 training steps, and kept the remaining setup the same as in [1]. At test time, we generated 100 clean images for forgery and measured the extracted watermark results (Det. / Acc.):
>
> |Method|Det.|Acc.|
> |-|-|-|
> |TR|0.66|-|
> |TR-S|0.02|-|
> |GS|0.13|0.6742|
> |GS-S|0.00|0.5495|
>
> The near-zero detection and accuracy values of TR-S and GS-S under WMCopier show that SemBind substantially improves resistance to this forgery attack. A likely reason is that, under a fixed key, TR and GS induce relatively fixed latent distributions, whereas SemBind binds the watermark to the prompt and thus produces much more diverse latent distributions, making them harder to imitate.
>
> [1] *Dong, Ziping, et al. "WMCopier: Forging Invisible Watermarks on Arbitrary Images." The Thirty-ninth Annual Conference on NeurIPS.*
>
> **2. Answer to W2&Q1 (Threat model is underspecified).**
>
> In our threat model, the attacker has access to a watermarked image dataset, can train surrogate models, and knows the public model architecture and algorithmic design, but has no access to detector queries, which we consider too strong.
>
> For the two possible attack directions:
>
> 1. Training a surrogate masker.
>    We additionally evaluated this attack under a favorable setting where the attacker knows the masker architecture, training data, and hyperparameters, and differs from the defender essentially only in the random seed. Even in this case, the attacker fails to train an effective surrogate masker due to the large solution space of the training objective. See **Rebuttal for Review dZcX, Answer to W2** for details.
>
> 2. Targeted optimization toward a desired latent mask.
>    In our setting, detector queries and detector feedback are unavailable to the attacker, making direct optimization toward a desired latent mask impractical.
>
> **3. Answer to W3 (Semantic-preserving edits).**
>
> We would like to clarify the distinction between robustness and anti-forgery. Images with only natural perturbations or minor benign modifications should be treated as robustness cases rather than forgeries. In realistic attacks, the goal is typically to transfer the provider’s watermark to semantically different, potentially harmful or policy-violating content, which is exactly the scenario SemBind is designed to defend against.
>
> We additionally conducted a prompt-level adaptive attack. In brief, SemBind remains tolerant to near-semantic-preserving prompt variations, but becomes increasingly discriminative once semantically different or unsafe content is injected. For details, please refer to **Rebuttal for Review P4WR, Answer to Q1**.
>
> **4. Answer to W4 (Mask ratio parameter experiment).**
>
> We conducted robustness and imprinting forgery experiments to study mask ratio selection.
>
> **Robustness on SDP**
>
> *(Same setting as Page 8 of the main paper; reporting average Dec. / Acc. over None, JPEG (QF=70), Brightness, GauBlur, GauNoise, MedFilter, and Resize.)*
>
> |Method| ratio = 1.0 | ratio = 0.75 | ratio = 0.5 | ratio = 0.25 |
> |-|-|-|-|-|
> |TR-S|0.992 / - |1.0 / - | 1.0 / -| 1.0 / -|
> |GS-S|0.993 / 0.988 |1.0 / 0.995| 1.0 / 0.998 | 1.0 / 1.0 |
> |PRC-S|0.752 / 0.882 |0.884 / 0.938| 0.940 / 0.969 | 0.982 / 0.990 |
> |GS++-S|0.786 / 0.896 |0.887 / 0.942| 0.932 / 0.958 | 0.965 / 0.984 |
>
> **Imprinting forgery results on SDP**
>
> *(SD2.1 as proxy model; same setting as Page 7; reporting Dec. / Acc. at Step 150.)*
>
> | Method | ratio = 1.0 | ratio = 0.75 | ratio = 0.5 | ratio = 0.25 |
> |-|-|-|-|-|
> | TR-S | 0.02 / -| 0.16 / - | 0.79 / - | 0.98 / - |
> | GS-S | 0.07 / 0.493 | 0.27 / 0.764 | 0.92 / 0.952 | 1.0 / 0.988 |
> | PRC-S | 0.02 / 0.498 | 0.08 / 0.541 | 0.25 / 0.506 | 0.54 / 0.781 |
> | GS++-S | 0.03 / 0.516 | 0.10 / 0.608 | 0.49 / 0.729 | 0.58 / 0.802 |
>
> The experimental results show that choosing ratio = 1.0 for TR-S and GS-S, and ratio = 0.5 for PRC-S and GS++-S, provides a better balance between robustness and anti-forgery performance.
>
> **5. Answer to W5 (Quantitative link, collision rate and tolerance thresholds).**
>
> For GS-S, the relation between mask mismatch and verification failure is as follows: accuracy drops by 0.3%, 0.8%, 3.7%, and 16.5% at mismatch levels of 10%, 20%, 30%, and 40%, respectively. At 50% mismatch, verification is close to random guessing.
>
> Our collision rate is very low. Even with a threshold of 80 bits, it is only 0.42% on COCO and 0.10% on SDP; please refer to **Review P4WR, Answer to Q1** for details.
>
> The effect of tolerance thresholds on collisions is beyond the scope of the current paper.

---

> > ### Author Rebuttal · Reviewer_Hcer · 2026-04-04
> >
> > Your explanation clarified my concerns, and I will be updating my rating accordingly. I hope the authors can include the missing experiments in the final version, and also consider improving the related work section as well as the overall formatting of the paper. （See Key Questions）

---

> > > ### Author Response · Authors · 2026-04-04
> > >
> > > Thank you for the thoughtful follow-up and for updating your rating. We are glad that our rebuttal helped clarify your concerns. We will incorporate the missing experiments, improve the related work discussion, and revise the overall formatting in the final version. Thank you again for your constructive feedback.

---

### Official Review · Reviewer_dZcX · 2026-03-11

**Soundness:** 3
**Presentation:** 2
**Significance:** 3
**Originality:** 2
**Overall Recommendation:** 4
**Confidence:** 3

**Summary:**

This paper proposes a new defense that binds the watermark to the semantic content of the image against watermark forgery attack. The method introduces a semantic masker that generates a prompt-dependent binary code, which is expanded into a sign mask applied to the watermarked latent before diffusion sampling. This design ensures that watermark detection depends not only on the latent watermark but also on semantic consistency. The method is designed to be compatible with existing latent watermarking methods, and experiments demonstrate improved resistance to watermark forgery attacks while maintaining image quality and most robustness properties of the original watermarking schemes.

**Compliance With Llm Reviewing Policy:**

Affirmed.

**Final Justification:**

The authors' rebuttal addresses most of my concern. I will keep the positive score.

**Key Questions For Authors:**

Please see weaknesses.

**Limitations:**

yes

**Strengths And Weaknesses:**

Strengths:
1. The paper studies a defense against watermark forgery attack, which is a realistic threat to latent diffusion watermarking systems.
2. The proposed method is designed as a wrapper that can be applied to multiple watermarking methods rather than a scheme-specific solution.
3. The idea of binding watermark recovery to image semantics is a natural and practical defense mechanism.

Weaknesses:
1. The experiments appear to rely on older Stable Diffusion models. It is unclear whether the method remains effective on more recent models such as Flux or newer diffusion architectures, which may have different latent properties.
2. The evaluation focuses mainly on imprinting and reprompting under relatively simple black-box assumptions. While these attacks are meaningful, the paper provides limited analysis of stronger adaptive attackers who may attempt to optimize prompts, query the system repeatedly, or learn surrogate models of the semantic masking mechanism. Since the defense relies on semantic binding and a learned masking function, a more thorough investigation of adaptive strategies would help clarify the true security guarantees of the method.
3. The security of the approach partially relies on the secrecy and robustness of the learned masker, which may introduce new attack surfaces. I think the author should analyze them.
4. How sensitive are the results to the mask ratio parameter? Providing a more detailed ablation would clarify the robustness security tradeoff.

---

> ### Author Rebuttal · Authors · 2026-03-30
>
> Thank you for the positive assessment and the constructive questions. We address the reviewer’s key concerns below.
>
> **1. Answer to W1 (New diffusion architectures such as FLUX).**
>
> We thank the reviewer for raising this point. In fact, we have already evaluated SemBind on the FLUX model mentioned by the reviewer in **Appendix E.2**. The results show that SemBind can be effectively integrated into FLUX and can continue to provide strong protection against forgery attacks; in our experiments, its defense performance is even stronger than on Stable Diffusion 2.1, while maintaining good robustness. Therefore, we believe SemBind is not restricted to older Stable Diffusion models and shows promising generalization to newer diffusion architectures.
>
> **2. Answer to W2 (Adaptive attack experiments).**
>
> In fact, we have already analyzed two representative adaptive attacks (surrogate semantic masker attack and pixel-space spoofing attack) in **Appendix D**, and during the rebuttal period we further conducted a prompt-level adaptive attack. Overall, these results suggest that SemBind remains resistant to these attack strategies.
>
> **(1) Surrogate semantic masker attack.**
> In this setting, the attacker knows the masker architecture, training data, and training hyperparameters, and differs from the defender essentially only in the random seed. This is already a stronger assumption than most realistic scenarios. Even under this favorable setting, our results show that training an effective surrogate masker remains highly challenging because the objective is defined over relative relationships rather than absolute targets.This leads to a very large solution space with many equivalent optima, causing the training outcomes to vary substantially across different random seeds.
>
> To further evaluate this attack path, during the rebuttal period we directly used the five surrogate maskers $M_1$–$M_5$, trained with different random seeds as described in Appendix D, to perform actual forgery attempts. Specifically, the attacker used the surrogate masker to bind its own watermark and then tested whether the resulting images could fool the verifier. Taking **GS-S** as an example, the extracted results (Det. / Acc.) were as follows:
>
> | Surrogate masker | Det. | Acc. |
> |-|-|-|
> |$M_1$| 0.00 | 0.5104 |
> |$M_2$| 0.00 | 0.5072 |
> |$M_3$| 0.00 | 0.4988 |
> |$M_4$| 0.00 | 0.5046 |
> |$M_5$| 0.00 | 0.5089 |
>
> These results show that the learned surrogate masker cannot substitute for the real masker in the forgery attack. In our experiments, this attack path appears to be highly ineffective and impractical.
>
> **(2) Pixel-space spoofing attack.**
> We also evaluated a pixel-space spoofing strategy, where the attacker progressively mixes the source watermarked image into the target image in order to drive the target semantic code toward that of the source image. The results show that the semantic code only becomes noticeably close to the source image when the source image covers **more than 80%** of the target image. This means that, in order to spoof the semantic masker successfully in pixel space, the attacker must introduce extremely strong and visually obvious perturbations, which substantially distort the content and naturalness of the target image. This greatly limits the practicality of such an attack.
>
> **(3) Prompt-level adaptive attack.**
> To further enrich the discussion of adaptive attacks, during the rebuttal period we additionally conducted a prompt-level adaptive attack study. The results show that SemBind behaves as intended: it is comparatively tolerant to near-semantic-preserving prompt variations, but becomes strongly discriminative once unsafe or out-of-intent content is injected. Please see **Rebuttal for Reviewer P4WR, Answer to Q1** for the detailed experimental setup and results.
>
>
> **3. Answer to W3 (New attack surfaces discussion).**
>
> We agree that SemBind relies on the semantic masker remaining private. If an attacker obtains the deployed masker parameters and can directly execute the bind/unbind operations, the anti-forgery mechanism is no longer effective. This white-box setting is outside the scope of our work.
>
> Under this threat model, we have already analyzed three relevant attack surfaces in our **Answer to W2**. Our results suggest that these attacks provide only limited practical advantage to a black-box attacker, while the gain in forgery resistance is significant. We will revise the paper to make this threat model and its limitations much more explicit.
>
> **4. Answer to W4 (Mask ratio parameter experiment).**
>
> Thank you for your suggestion. We conducted experiments on robustness and anti-forgery performance under different mask ratios. The results show that choosing a ratio of 1.0 for TR-S and GS-S, and 0.5 for PRC-S and GS++-S, provides a better balance between robustness and anti-forgery capability.
> For details, please see **Rebuttal for Reviewer Hcer, Answer to W4**.

---

> > ### Author Rebuttal · Reviewer_dZcX · 2026-04-03
> >
> > Thanks for your rebuttal. I will keep my positive score.

---

> > > ### Author Response · Authors · 2026-04-04
> > >
> > > Thank you for your positive feedback and for keeping your positive score. We appreciate your time and consideration.

---

### Official Review · Reviewer_P4WR · 2026-03-12

**Soundness:** 3
**Presentation:** 3
**Significance:** 3
**Originality:** 3
**Overall Recommendation:** 4
**Confidence:** 3

**Summary:**

The paper introduces SemBind, a method that improves the unforgeability of existing latent-space diffusion watermarks. Therefore SemBind operates in a two-stage process, where first a clean image is generated given an input prompt. Then a semantic masker extracts a binary message that encodes the semantic information of the image. In the second stage, a watermarked image is produced and the watermarked initial noise is combined with the semantic message. By combining the semantic information with the watermark information, the watermark can no longer be forged onto unrelated images.

**Compliance With Llm Reviewing Policy:**

Affirmed.

**Final Justification:**

The rebuttal has addressed my concerns and I will maintain my score. The paper provides clear merits, however the requirement to perform (at least) one additional generation cycle to provide forgery protection, limits its usability.

**Key Questions For Authors:**

1. How high is the collision rate of the semantic masker and is it possible for an attacker to forge the watermark, given watermarked image x, onto a visually similar image x’? For example images generated with the same or similar prompt across different seeds. Can this also result in failure cases when the two images that are generated in the two different stages of generateion for SemBind are too dissimilar?

2. Can SemBind be used as an independent watermark without using the existing watermarking methods?

3. How well does SemBind perform against advanced removal attacks such as CtrlRegen?

**Limitations:**

yes

**Strengths And Weaknesses:**

### Strengths:
- SemBind is compatible with existing watermarking schemes.
- The paper provides an extensive experimental evaluation across four latent-space diffusion watermarks and evaluating the unforgeability against two forging attacks.

### Weaknesses:
- To generate an image watermarked with SemBind, the generation process has to be performed twice, resulting in additional compute overhead.
- The robustness test of SemBind would be strengthened by also evaluating advanced attacks such as CtrlRegen [1].

### Minor Weakness
- The header for each page is currently “Submission and Formatting Instructions for ICML 2026”, which should be fixed for the final version.


**References**

[1] Yepeng Liu et al. “Image Watermarks are Removable using Controllable Regeneration from Clean Noise”. ICLR. 2025.

---

> ### Author Rebuttal · Authors · 2026-03-30
>
> Thank you for the positive assessment and the constructive questions. We address the reviewer’s key concerns below.
>
> **1. Answer to Q1 (Collision rate, similar image/prompt forgery, failure cases).**
>
> **(1) Collision rate**
>
> We further evaluated the collision rate of the semantic masker on COCO and SDP by randomly sampling 500 prompts, generating one image per prompt, extracting the corresponding 1024-bit semantic code, and computing all pairwise Hamming distances (HD) across different prompts. The resulting collision rates are very low across multiple HD thresholds: for HD < 20, the rates are 0.01% on COCO and 0.006% on SDP; for HD < 50, they are 0.09% on COCO and 0.04% on SDP; and even for the relatively loose threshold HD < 80, they remain only 0.42% on COCO and 0.10% on SDP. This suggests that semantic codes from different prompts are typically well separated, and accidental collisions remain rare even under a relaxed collision threshold.
>
> **(2) Similar image/prompt forgery**
>
> *Is it possible for an attacker to forge the watermark using a similar prompt or a similar image?*
>
> Yes, it is possible for an attacker to transfer the watermark using a similar prompt or a similar image. However, we view this case as closer to the **robustness boundary** than to the typical **forgery** setting. A key point here is that an image with only natural perturbations or minor benign modifications should not be treated as a forgery, but rather as a case of robustness. In realistic forgery attacks, the attacker usually aims to transfer the provider’s watermark to semantically different content (e.g., harmful or policy-violating content), which is exactly the scenario SemBind is designed to defend against.
>
>
> We further performed a **prompt-level adaptive attack** study. For each of COCO and SDP, we sampled 200 benign prompts, and we additionally used 200 unsafe prompts from I2P[1] to construct:
> - $P_0$: original benign prompt
> - $P_1$: a benign paraphrase or synonym-substituted version of $P_0$
> - $P_2$: $P_0$ plus half of the tokens from an I2P prompt
> - $P_3$: full I2P prompt
>
> The average semantic-code Hamming distances were:
>
> | Dataset | P0 → P1 | P0 → P2 | P0 → P3 |
> |-|-|-|-|
> | COCO    | 81.2     | 313.1    | 524.9    |
> | SDP     | 123.7    | 310.8    | 463.0    |
>
>
> Thus, SemBind behaves as intended: it is comparatively tolerant to near-semantic-preserving prompt variations, but becomes strongly discriminative (approaching 512 bits) once unsafe or out-of-intent content is injected.
>
> **(3) Failure cases**
>
> *Can this also lead to failure cases when the two images generated in the two stages of SemBind are too dissimilar?*
>
> Yes, this failure mode can occur when the two stages of SemBind generate images that are too dissimilar. A simple mitigation is to perform a self-check and re-run SemBind with a newly generated reference image. We agree that SemBind introduces additional computational overhead, but we believe the resulting anti-forgery capability is well worth the cost.
>
> **2. Answer to Q2 (Can SemBind be used as an independent watermarking method?).**
>
> No. SemBind is not designed as an independent watermarking scheme; rather, it serves as a semantic-binding enhancement built on top of existing latent-space watermarking methods and is broadly applicable to current latent-based watermarking approaches.
>
> **3. Answer to Q3 (Removal attacks experiment).**
>
> Thank you for the suggestion. We additionally evaluated CtrlRegen[2] on COCO using Stable Diffusion 2.1 Base, following the same settings as in the paper. We tested Tree-Ring (TR) and Gaussian Shading (GS), together with their SemBind variants (TR-S and GS-S), on 100 prompts.
>
> The results (Det. / Acc.) before and after the CtrlRegen attack are as follows:
>
> | Method | Before CtrlRegen (Det. / Acc.) | After CtrlRegen (Det. / Acc.) |
> |--------|--------------------------------|-------------------------------|
> | TR     | 1.00 / -                       | 0.23 / -                      |
> | TR-S   | 1.00 / -                       | 0.22 / -                      |
> | GS     | 1.00 / 1.000                   | 0.98 / 0.882                  |
> | GS-S   | 0.99 / 0.998                   | 0.96 / 0.879                  |
>
> Thus, the removal performance of CtrlRegen on the SemBind variants is very close to that on the corresponding base schemes. This is because CtrlRegen aims to preserve the image semantics while removing the watermark, so the recomputed semantic mask remains similar.
>
> [1] *https://huggingface.co/datasets/AIML-TUDA/i2p*
>
> [2] *Y. Liu, Y. Song, H. Ci, et al. Image Watermarks are Removable using Controllable Regeneration from Clean Noise. ICLR 2025.*

---

> > ### Author Rebuttal · Reviewer_P4WR · 2026-04-02
> >
> > I would like to thank the authors for their efforts in the rebuttal and their detailed response.
> >
> > The experiments on the collision rate and prompt forgery have been convincing. However, the failure case resolvement per  self-check and re-running SemBind further increases the computational overhead of the method and should be addressed in the Limitations.
> >
> > The additional robustness experiments show that Sem-Bind does not impact the robustness negatively, even under advanced removal attacks.
> >
> > While the paper provides clear merits, the requirement to perform (at least) one additional generation cycle to provide forgery protection, limits its usability. Therefore, I would like to maintain my score.

---

> > > ### Author Response · Authors · 2026-04-04
> > >
> > > Thank you very much for your positive assessment. We also agree that the self-check and possible re-generation introduce additional computational overhead, which may limit practical usability. We will explicitly discuss this trade-off in the Limitations section of the final version. Thank you again for your constructive feedback.

---

### Decision · Program_Chairs · 2026-04-30

**Decision:**

Accept (regular)

**Comment:**

This submission proposes SemBind, which is a defense for latent-based diffusion watermarks against black-box forgery attacks. The method binds latent signals to image semantics via a learned semantic masker. This masker is trained using contrastive learning.

SemBind operates in a two-stage process:
1. A clean image is generated given an input prompt. Then a semantic masker extracts a binary message that encodes the semantic information of the image.
2. The watermarked initial noise is combined with the semantic message to produce a watermarked image.

By combining the semantic information with the watermark information, the watermark is protected from being easily forged onto unrelated images.

All the Reviewers unanimously agree: "Weak Accept":
1. Reviewer P4WR indicates that a big plus is that SemBind is compatible with existing watermarking schemes and thoroughly evaluated. The drawback of the method is that the generation has to be performed twice for the clean and the watermarked image, which adds > 2X overhead than for a standard generation, which limits its usability. Furthermore, the failure case resolvement per self-check and re-running SemBind further increases the computational overhead of the method and should be addressed in the Limitations. The formatting of the submission is incorrect, which should be improved for the final version of the paper.
2. Reviewer dZcX points out that if an attacker obtains the deployed masker parameters and can directly execute the bind/unbind operations, the anti-forgery mechanism is no longer effective. This should be clearly indicated in the limitations and more specific description of how this can be improved should be added there. Overall, the evaluation on Stable Diffusion, Flux, and for the adaptive attacks sufficiently addressed Reviewer's concerns.
3. As Reviewer Hcer stated, the authors should include the new experiments added during the rebuttal in the final version, and also consider improving the related work section as well as the overall formatting of the paper.
4. Reviewer m3qT also highlighted no measurable image quality degradation incurred by the method. The new experiments should be explicitly added by the authors, especially for the CtrlRegen (as also indicated by Reviewer P4WR).

Overall, I adhere to Reviewer's assessment and mark the paper as accepted, conditioned on the above points raised by the Reviewers: the improvements have to be integrated into the camera-ready version of the submission.